# Defining a TCF1-expressing progenitor allogeneic CD8+ T cell subset in acute graft-versus-host disease

Solhwi Lee[1,5], Kunhee Lee[1,5], Hyeonjin Bae[1], Kyungmin Lee[1], Junghwa Lee[2], Junhui Ma[1], Ye Ji Lee[3], Bo Ryeong Lee[3], Woong-Yang Park[3,4] & Se Jin Im[1] ✉

Graft-versus-host disease (GvHD) is a severe complication of hematopoietic stem cell transplantation driven by activated allogeneic T cells. Here, we identify a distinct subset of T cell factor-1 (TCF1)+ CD8+ T cells in mouse allogeneic and xenogeneic transplant models of acute GvHD. These TCF1+ cells exhibit distinct characteristics compared to TCF1- cells, including lower expression of inhibitory receptors and higher expression of costimulatory molecules. Notably, the TCF1+ subset displays exclusive proliferative potential and could differentiate into TCF1- effector cells upon antigenic stimulation. Pathway analyses support the role of TCF1+ and TCF1- subsets as resource cells and effector cells, respectively. Furthermore, the TCF1+ CD8+ T cell subset is primarily present in the spleen and exhibits a resident phenotype. These findings provide insight into the differentiation of allogeneic and xenogeneic CD8+ T cells and have implications for the development of immunotherapeutic strategies targeting acute GvHD.

Allogeneic hematopoietic stem cell transplantations (allo-HSCT) are used to treat hematological malignancies and genetic diseases including leukemia, aplastic anemia, and bone marrow failure. In addition to reconstituting with functional donor-derived immune cells, allo-HSCT results in the activation of alloreactive T cells upon recognition of host major histocompatibility complex (MHC) molecules, contributing to the graft-versus-tumor (GvT) effect in patients with hematological malignancies[1]. However, activated alloreactive T cells also attack the host tissues and lead to graft-versus-host disease (GvHD), which is a major cause of mortality in patients who receive allo-HSCT[2–6]. Approximately 10% to 50% of patients who receive MHC-matched transplantation of T cells die because of GvHD-related complications despite pharmacology prophylaxis and treatments[6–8]. Cytotoxic CD8+ T cells play a pivotal role in the pathogenesis of acute GvHD because they directly attack nonmalignant host tissues through effector molecules[9–11]. However, the details of the differentiation pathways of alloreactive CD8+ T cells in acute GvHD are not fully understood.

One of the representative features of acute GvHD is that T cells are exposed to persistent antigenic stimulation, similar to conditions found in chronic viral infections, the tumor microenvironment, and autoimmune diseases. Using a mouse model of chronic lymphocytic choriomeningitis virus (LCMV) infection, we discovered a TCF1+CXCR5+ progenitor exhausted CD8+ T cell subset[12]. The TCF1+PD-1+ progenitor CD8+ T cells act as resource cells that maintain a pool of antigen-specific CD8+ T cells and give rise to Tim-3+PD-1+ terminally differentiated CD8+ T cells, which retain effector functions, by antigen-driven proliferation and differentiation[12–15]. More importantly, these TCF1+PD-1+ CD8+ T cells exclusively exhibit a proliferative burst in response to PD-1-targeted therapy. The concept of the progenitor-progeny relationship of antigen-specific T cells extends to the tumor microenvironment[16–20] and autoimmune diseases[21,22]. In both mouse and human tumor models, the TCF1+PD-1+ CD8+ T cell subset exhibited high proliferative potential, a better efficacy in controlling tumors, and superior therapeutic effects combined with a PD-1 blockade compared

[1]Department of Immunology, Graduate School of Basic Medical Science, Sungkyunkwan University School of Medicine, Suwon, Republic of Korea. [2]Department of Precision Medicine, Sungkyunkwan University School of Medicine, Suwon, Republic of Korea. [3]GENINUS Inc., Seoul, Republic of Korea. [4]Samsung Genome Institute, Samsung Medical Center, Seoul, Republic of Korea. [5]These authors contributed equally: Solhwi Lee, Kunhee Lee. ✉e-mail: sejinim@skku.edu

to the TCF1⁻ subset[18]. Notably, TCF1⁺PD-1⁺ CD8⁺ T cell subsets were mainly present in the lymphoid organs including the spleen[12,13] and tertiary lymphoid organs[16] in chronic infection and cancer, respectively. Furthermore, antigen-specific exhausted CD8⁺ T cells were resident in these models and only a small fraction of exhausted cells could circulate[16,23]. In the context of CD4⁺ T cell differentiation in autoimmune diseases, such as inflammatory bowel disease and experimental autoimmune encephalomyelitis, TCF1⁺ CD4⁺ T cells were quiescent and could replenish TCF1⁻ CD4⁺ T cells, which display effector functions in pathogenesis[21,22]. Thus, generation of the TCF1⁺ progenitor T cell subset would be a universal T cell differentiation program in the microenvironment with continuous antigenic stimulation.

In this work, given that alloreactive CD8⁺ T cells are also exposed to persistent antigenic stimulation and a high antigen load, we conduct a direct comparison of phenotypic, transcriptional, and functional characteristics of PD-1⁺ alloreactive CD8⁺ T cells in mice with MHC-mismatched acute GvHD to the corresponding subsets in chronically LCMV-infected mice. We identify the TCF1⁺ progenitor subset among PD-1⁺ alloreactive CD8⁺ T cells and discover similarities in the phenotypes, proliferative potential, and gene signatures between the two models. We also observe the presence of the TCF1⁺PD-1⁺ CD8⁺ T cell subset in the MHC-matched allogeneic GvHD and xenogeneic GvHD models. In addition, the TCF1⁺ progenitor CD8⁺ T cell subset predominantly localizes in the spleen and exhibits better responsiveness to IL-15 treatment ex vivo. Thus, these data provide new insight into T cell–mediated pathogenesis in acute GvHD and novel perspectives for developing new immunotherapeutics.

## Results

### Generation of the TCF1⁺ subset of PD-1⁺ alloreactive CD8⁺ T cells in acute GvHD

To determine whether the TCF1⁺ subset is generated among alloreactive CD8⁺ T cells during the development of acute GvHD, we transplanted total splenocytes and bone marrow cells isolated from Balb/c (H2ᵈ) mice into lethally irradiated C57bl/6 (B6) recipient mice (H2ᵇ) to induce GvHD (Fig. 1A–C). For the parallel comparison, we chronically infected B6 mice with LCMV clone 13 (Cl-13) strain after a transient depletion of CD4⁺ T cells to induce lifelong viremia[24]. We first found that most CD8⁺ T cells in the spleen of mice with acute GvHD at 7 days post-transplantation (dpt) were composed of donor T cells (Supplementary Fig. 1) and ~90% of CD8⁺ T cells highly expressed PD-1, reflecting TCR-mediated activation of alloreactive T cells (Fig. 1D). Furthermore, as we hypothesized, a distinct TCF1⁺Tim-3⁻ CD8⁺ T cell subset was generated during the evolution of acute GvHD, and the frequency of TCF1⁺ cells among alloreactive PD-1⁺ CD8⁺ T cells was almost identical to that in chronically LCMV-infected mice. Phenotypic characteristics of PD-1⁺ CD8⁺ T cells were overall similar between acute GvHD and chronic LCMV infection (Fig. 1E–I and Supplementary Fig. 2). Although TCF1⁺ alloreactive CD8⁺ T cells exhibited lower expression of Tim-3 and CD39, they showed comparable TIGIT and higher PD-1 expression than TCF1⁻ CD8⁺ T cells. Additionally, TCF1⁺ CD8⁺ T cells showed higher expression of costimulatory molecules including CD28 and ICOS, as well as activation marker CXCR3, and lower expression of effector marker granzyme B. Both TCF1⁺ and TCF1⁻ subsets of PD-1⁺ alloreactive CD8⁺ T cells in the acute GvHD mouse model expressed CD44 and Eomes similarly, but TCF1⁺ cells presented higher TOX and lower T-bet expression than TCF1⁻ cells. Consistent with the notion that PD-1-expressing cells are predominantly LCMV-specific in the context of chronic LCMV infection, we also confirmed a similar differentiation pattern of LCMV tetramer-positive CD8⁺ T cells to PD-1⁺ CD8⁺ T cells in the spleen at 7 days post-infection (dpi) (Supplementary Fig. 3).

We next examined the maintenance of the TCF1⁺ CD8⁺ T cell subset after allogeneic transplantation. The frequency of TCF1⁺Tim-3⁻ cells among PD-1⁺ CD8⁺ T cells was comparable between 7 and 28 dpt

(Fig. 2A, B). However, the absolute number of alloreactive TCF1⁺PD-1⁺ CD8⁺ T cells significantly declined over time in the spleen (Fig. 2B). This is associated with the remarkably reduced number of PD-1⁺ CD8⁺ T cells in addition to total splenocytes and CD8⁺ T cells despite the augmented frequency of CD8⁺ T cells at 28 dpt (Supplementary Fig. 4). Similar to the results at 7 dpt, TCF1⁺PD-1⁺ CD8⁺ T cells exhibited lower expression of inhibitory receptors and higher expression of costimulatory molecules and activation markers than the TCF1⁻ subset. Low expression of Tox, Eomes, and T-bet in addition to granzyme B was observed in the TCF1⁺ CD8⁺ T cell subset at 28 dpt (Supplementary Fig. 5). Notably, the expression pattern of PD-1 and TOX in TCF1⁺ and TCF1⁻ CD8⁺ T cells during acute GvHD was the opposite between 7 and 28 dpt (Figs. 1E, H, and Supplementary Fig. 5), and this phenomenon was also observed in chronically infected mice (Figs. 1E, H, and Supplementary Fig. 6). Different from chronically infected mice (Supplementary Fig. 6), there was a considerable population of TCF1⁻Tim-3⁻ PD-1⁺ CD8⁺ T cells in mice with acute GvHD at 28 dpt (Fig. 2A). TCF1⁻Tim-3⁻ CD8⁺ T cells exhibited a trend of intermediate differentiation between TCF1⁺Tim-3⁻ and TCF1⁻Tim-3⁺ CD8⁺ T cells (Supplementary Fig. 5).

To examine the proliferation of CD8⁺ T cell subsets, we analyzed intracellular Ki-67 expression during acute GvHD and chronic LCMV infection. In both models, PD-1⁺ CD8⁺ T cells highly expressed Ki-67, suggesting they were in active phases of the cell cycle[25], at day 7 post-transplantation or post-infection (Fig. 2C). Over time, however, the TCF1⁺PD-1⁺ CD8⁺ T cell subset became quiescent and only approximately half of the TCF1⁻ cells showed Ki-67 expression. Next, we investigated CD101 expression, which marks terminally exhausted CD8⁺ T cells during chronic LCMV infection[26]. Consistent with the results in chronically infected mice, CD101 was absent in PD-1⁺ CD8⁺ T cells at 7 dpt, but the upregulation of CD101 was observed exclusively in the TCF1⁻ CD8⁺ T cell subset in acute GvHD (Fig. 2D). Finally, we determined the status of memory markers including CD127 (IL-7R alpha) and CD62L. In contrast to exhausted CD8⁺ T cells in chronically infected mice that did not express both molecules, most TCF1⁺PD-1⁺ CD8⁺ T cells in mice with acute GvHD highly expressed CD127 although they still lacked CD62L expression (Fig. 2E). Taken together, these results suggest that the characteristics of PD-1⁺ CD8⁺ T cells were quite similar between acute GvHD and chronic LCMV infection, and the generation of the TCF1⁺PD-1⁺ CD8⁺ T cell subset is a general adaptation of CD8⁺ T cell differentiation in a microenvironment with persistent antigenic stimulation.

### Antigen-driven CD8⁺ T cell differentiation and its PD-1 upregulation in allogeneic and xenogeneic GvHD models

In the current study, we considered PD-1⁺ CD8⁺ T cells as activated alloreactive T cells based on the fact that TCR-mediated signals upregulate PD-1. However, besides antigenic stimulation, it has been suggested that homeostatic proliferation and cytokines could also upregulate PD-1 on T cells[27,28]. To investigate whether proinflammatory cytokines and homeostatic proliferation of CD8⁺ T cells upregulate PD-1 expression, we assessed this phenomenon using a syngeneic transplantation model after irradiation to deplete host immune cells, which leads to lymphopenia and the production of pro-inflammatory cytokines. We isolated congenically distinct B6 splenocytes (CD45.1/CD45.1) and bone marrow cells (CD45.1/CD45.2) and transferred them into CD45.2 B6 recipients (Fig. 3A). After seven days, we examined the composition of CD8⁺ T cells in the spleen. The results showed that ~90% of CD8⁺ T cells were derived from donor splenocytes, while around 7% and 3% were derived from donor bone marrow cells and residual host cells, respectively (Fig. 3B, C). Interestingly, in contrast to allogeneic transplantation, although most CD8⁺ T cells highly expressed Ki-67, reflecting their homeostatic proliferation, only 10% of splenocyte-derived CD8⁺ T cells expressed PD-1 and most of these cells exhibited a phenotype of TCF1⁺TOX⁻Tim-3⁻ (Fig. 3D, E), similar to naïve CD8⁺ T cells (Supplementary Fig. 7). These findings

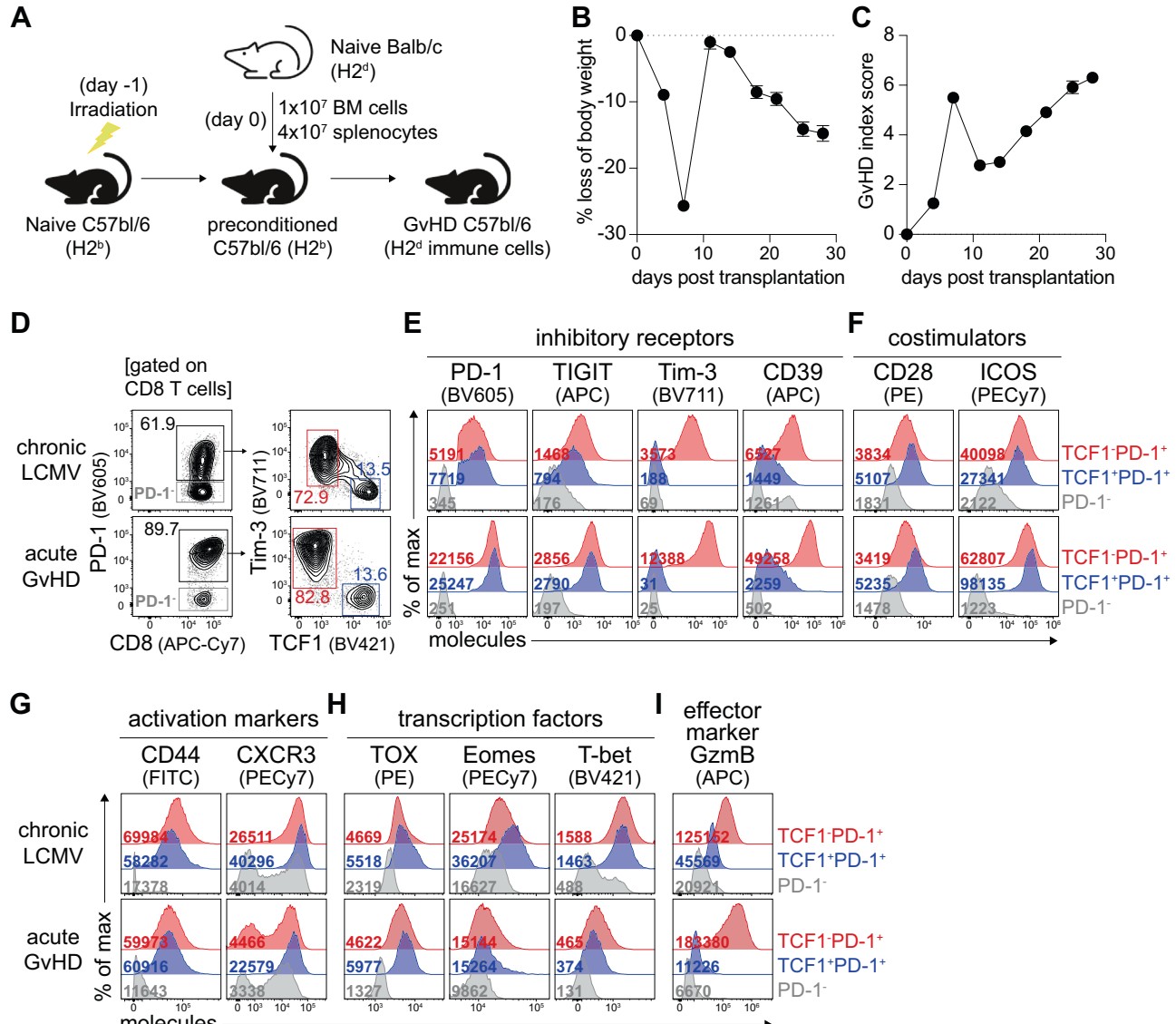

**Fig. 1 | Generation of the TCF1+ subset of PD-1+ alloreactive CD8+ T cells in acute GvHD. A** Experimental setup: One day after a lethal dose (900 cGy) of irradiation, B6 mice received $1 \times 10^7$ bone marrow cells and $4 \times 10^7$ splenocytes isolated from naïve Balb/c mice intravenously. **B**, **C** Kinetics of body weight loss (**B**) and GvHD index score (**C**) during the course of acute GvHD. Data were combined from four independent experiments ($n = 81$), and the mean and SEM were shown. **D–I** In parallel with acute GvHD, B6 mice were chronically infected with LCMV Cl-13 virus after transient depletion of CD4+ T cells. **D** Gating strategy to determine TCF1+PD-1+ and TCF1-PD-1+ CD8+ T cell subsets as well as PD-1- CD8+ T cells in the spleen of chronically infected mice (top) and mice with acute GvHD (bottom) at 7 days post-infection or post-transplantation. **E–I** Phenotypic analysis of PD-1+ CD8+ T cell subsets during chronic viral infection and acute GvHD. Representative flow plots show the expression of inhibitory receptors (**E**), costimulators (**F**), activation markers (**G**), transcription factors (**H**), and effector marker granzyme B (**I**) on the indicated CD8+ T cell subset in the two models. The numbers in the graph represent the mean fluorescence intensity. Data are representative of two independent experiments, $n = 4$–5 mice per group per experiment. Source data are provided as a Source Data file.

suggest that PD-1 upregulation occurs primarily due to allogeneic stimulation rather than inflammation and homeostatic proliferation after conditioning.

To determine the clinical relevance of our observations, we next examined the differentiation of CD8+ T cells in a bone marrow transplantation model that was MHC-matched (H2b), but had multiple minor antigen mismatches (129/Sv to B6) (Fig. 3F). After seven days, ~40% of splenic CD8+ T cells were PD-1-positive (Fig. 3G, H), which was significantly lower than the frequency observed in the MHC-mismatched model (90%; Balb/c to B6, Fig. 1D and Supplementary Fig. 4C). We then compared the phenotype of splenic CD8+ T cells based on PD-1 expression. PD-1-CD44lo CD8+ T cells exhibited a TCF1+Tim-3-TOX- phenotype, while PD-1+CD44hi CD8+ T cells were divided into TCF1+Tim-3- and TCF1-Tim-3+ subsets, and both subsets

showed high expression of TOX compared to PD-1-CD44lo CD8+ T cells (Fig. 3H, I). Combining these results with those of the MHC-mismatched model, we propose that the generation of TCF1+ and Tim-3+ CD8+ T cell subsets represents a general differentiation program during alloreactive T cell activation.

Next, we assessed whether persistent antigenic stimulation of xenogeneic antigens leads to a similar differentiation program in human CD8+ T cells. To investigate this, we utilized the xenogeneic GvHD model by infusing human PBMCs into NOD/LtSz-Prkdcscidil2rγtm1Wjl (NSG) mice (Fig. 3J). The infused human PBMCs (hPBMCs) which exhibit the strongest mixed lymphocyte reaction (MLR) to NSG splenocytes were selected for the experiment (Supplementary Fig. 8A), and we examined CD8+ T cell responses in the spleen on day 21 post-infusion when a significant decrease in body

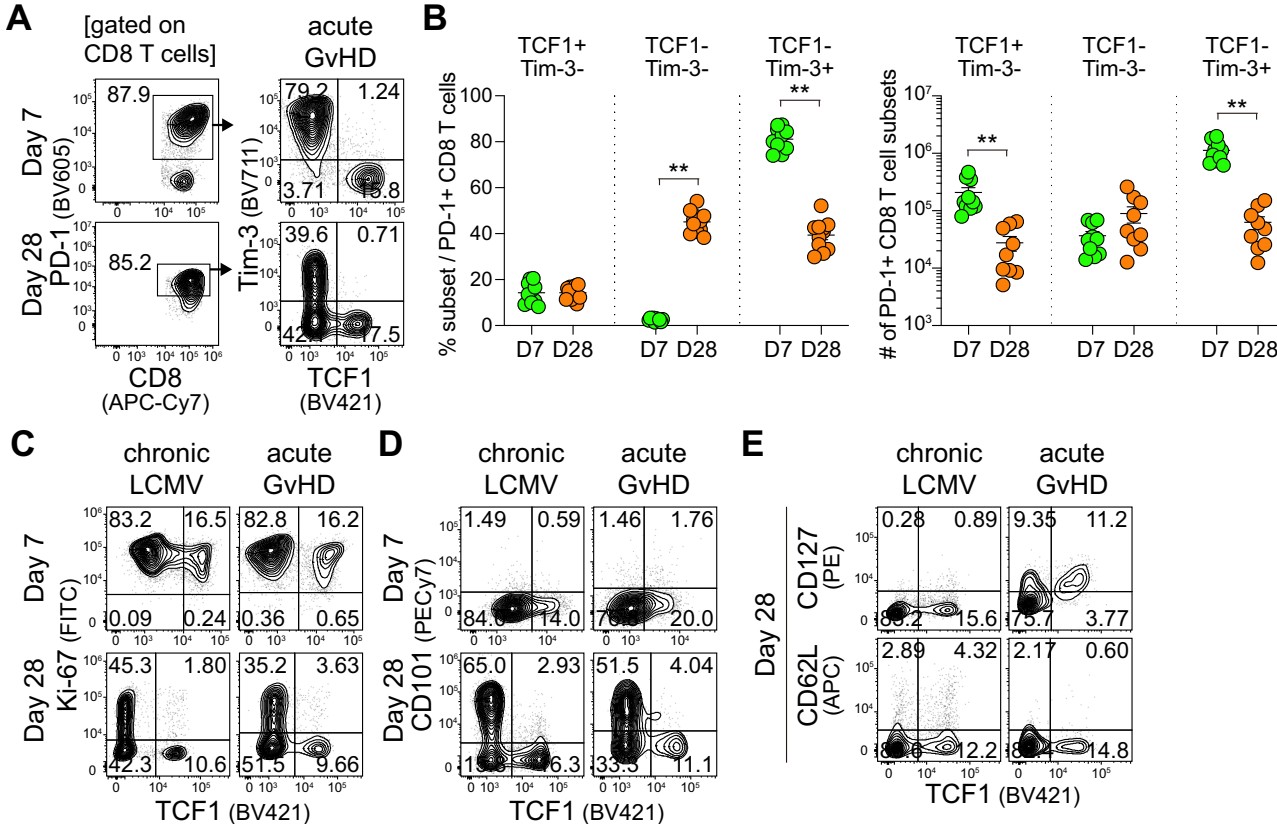

**Fig. 2 | Quiescent, less differentiated, and memory-like features of the TCF1⁺ CD8⁺ T cell subset in acute GvHD. A, B** Representative flow plots (**A**) and summary graphs (**B**) showing the frequency (left) and absolute number (right) of TCF1⁺Tim-3⁻, TCF1⁻Tim-3⁻, and TCF1⁻Tim-3⁺ subsets among PD-1⁺ CD8⁺ T cells at day 7 and 28 post-transplantation in the spleen of mice with acute GvHD. Data are combined from two independent experiments (n = 9 for day 7 and n = 10 for day 28), and the mean and SEM were shown. Statistical significance was determined by a two-tailed unpaired t-test. Source data are provided as a Source Data file. **C, D** TCF1 vs. Ki-67 (**C**) or CD101 (**D**) expression on PD-1⁺ CD8⁺ T cells in the spleen at the indicated time points. **E** Expression of memory markers, CD127 and CD62L, vs. TCF1 in PD-1⁺ CD8⁺ T cells in the spleen at 28 dpt.

weight and an increase in GvHD index score were observed (Supplement Fig. 8B, C). Similar to the findings in the MHC-mismatched allogeneic GvHD model (Fig. 1B), we observed that ~90% of CD8⁺ T cells expressed PD-1 (Fig. 3K, L). Furthermore, distinct subsets of TCF1⁺ and Tim-3⁺ cells were observed, accompanied by high TOX expression (Fig. 3L, M). We further characterized the xeno-reactive PD-1⁺ CD8⁺ T cells based on their CCR7 and CD45RA expression including naïve T cells (Tn, CCR7⁺CD45RA⁺), central memory T cells (Tcm, CCR7⁺CD45RA⁻), effector memory/effector T cells (Tem/eff, CCR7⁻CD45RA⁻), and CD45RA⁺ effector memory T cells (Temra, CCR7⁻CD45RA⁺). Intriguingly, the majority of TCF1⁻PD-1⁺ CD8⁺ T cells were Tem/Teff cells, while TCF1⁺PD-1⁺ CD8⁺ T cells consisted of both Tcm and Tem/Teff cells in equal proportions (Fig. 3N, O). These findings collectively indicate the generation of a TCF1⁺PD-1⁺ CD8⁺ T cell subset in response to both allogeneic and xenogeneic antigenic stimulation.

**Preferential localization of the TCF1⁺ PD-1⁺ alloreactive CD8⁺ T cell subset in the spleen during acute GvHD**

We previously demonstrated that TCF1⁺ progenitor exhausted CD8⁺ T cells were observed only in lymphoid organs such as the spleen, bone marrow, and lymph nodes of chronically LCMV-infected mice[12,13] and the tertiary lymphoid structures of patients with head and neck cancer[16]. In addition, although virus-specific exhausted CD8⁺ T cells recirculated at the early phase of the chronic infection, they became resident and only a small fraction of the transitory subset could be observed in the blood at the later phase of chronic infection[23]. Here we examined the population of TCF1⁺PD-1⁺ alloreactive CD8⁺ T cells in

the blood, liver, and spleen of mice during acute GvHD. Consistent with the results observed in chronically infected mice and cancer patients, the frequency of TCF1⁺Tim-3⁻ cells among PD-1⁺ CD8⁺ T cells was much higher in the spleen than that in the blood and liver at both time points, 7 and 28 dpt (Fig. 4A–D), suggesting the universal properties of TCF1⁺PD-1⁺ CD8⁺ T cells in their localization into the lymphoid tissues in the microenvironment with persistent antigenic stimulation. Furthermore, the frequency of CD8⁺ T cells among total lymphocytes at 28 dpt in the blood was 2% to 3% while that in the liver and spleen reached 40% and 30%, respectively. We also found that ~85% of CD8⁺ T cells (ranging from 75% to 95%) expressed PD-1 at 7 dpt in the blood, but the frequency dropped to 23% by 28 dpt, reflecting impaired circulation of alloreactive CD8⁺ T cells during acute GvHD. To further analyze their location in the spleen, we performed in vivo intravascular staining by injection of BV421-conjugated anti-CD45.2 antibodies[29]. We found that all circulating CD8⁺ T cells were labeled (Fig. 4E), and the proportion of antibody-labeled PD-1⁺ CD8⁺ T cells in the spleen was higher among TCF1⁻ cells than TCF1⁺ cells (Fig. 4F, G), suggesting that TCF1⁻ cells were more accessible to the blood in the red pulp. Consistent with these findings, we observed that more than half of the PD-1⁺ alloreactive CD8⁺ T cells expressed the resident marker CD69 and the TCF1⁺ CD8⁺ T cell subset showed higher levels of CD69 (Fig. 4H). Although we determined the alloreactive CD8⁺ T cells based on their PD-1 expression not using the tetramers, these results suggest that alloreactive CD8⁺ T cells become resident over time and the TCF1⁺ CD8⁺ T cell subset localizes in the spleen preferentially as observed in other models with continuous antigenic stimulants.

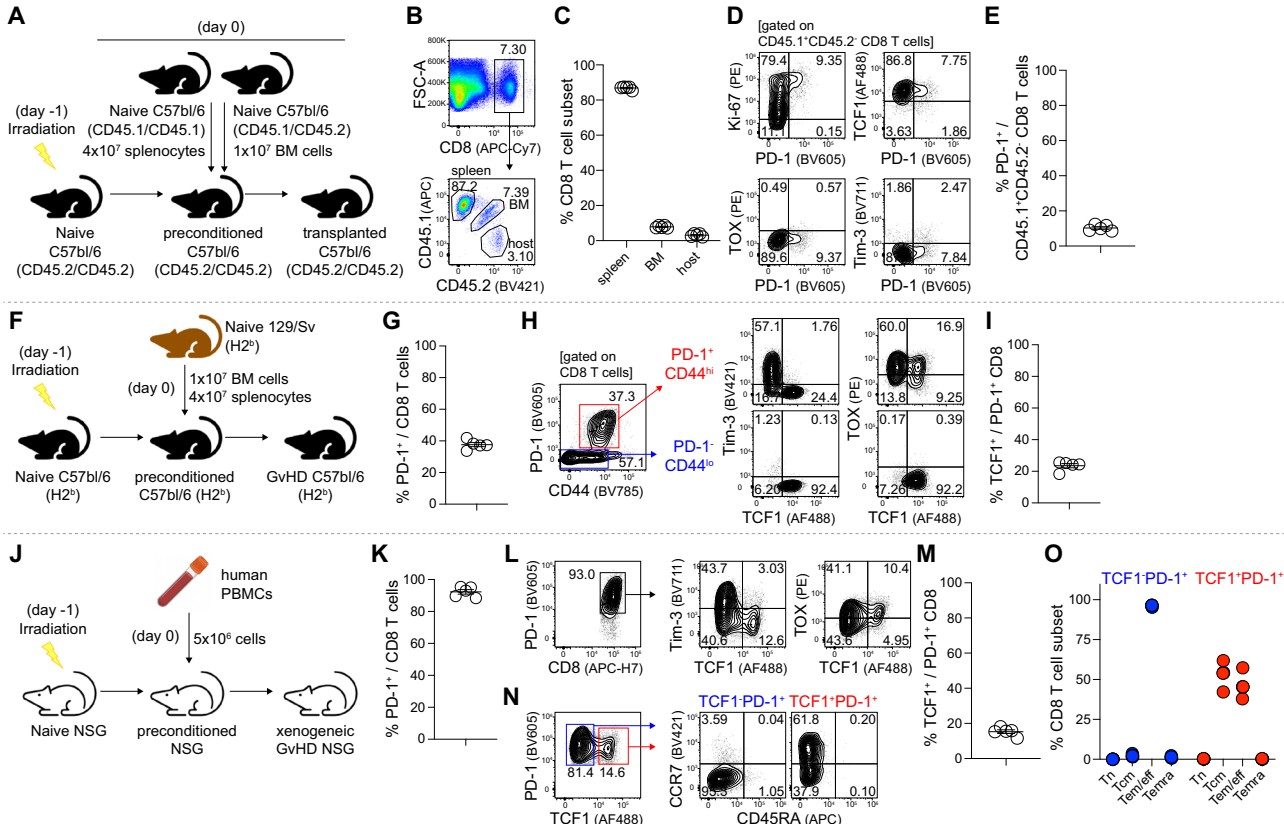

**Fig. 3 | Antigen-driven CD8⁺ T cell differentiation and PD-1 upregulation in allogeneic and xenogeneic GvHD models. A–E** Syngeneic transplantation model. **A** Experimental setup for the syngeneic transplantation. **B** Population of CD8⁺ T cells in the spleen at 7 dpt (top) and their origins (bottom). **C** Summary graph showing the population of CD8⁺ T cell subsets based on their origins. **D** Representative flow plots of the expression of PD-1 vs. Ki-67, TCF1, TOX, and Tim-3 among CD45.1⁺CD45.2⁻ CD8⁺ T cells. **E** Summary graph showing the population of PD-1⁺ cells among splenocyte-derived CD45.1⁺CD45.2⁻ CD8⁺ T cells. **F–I** An MHC-matched, minor antigen-mismatched allogeneic GvHD model. **F** Experimental setup for the MHC-matched, minor-antigen mismatched allogeneic transplantation. **G** Summary graph showing the population of PD-1⁺ cells among CD8⁺ T cells. **H** Representative flow plots of the population of PD-1⁺CD44ʰⁱ and PD-1⁻CD44ˡᵒ CD8⁺ T cell subsets and their TCF1, Tim-3, and TOX expression. **I** Summary graph

showing the population of the TCF1⁺ subset among PD-1⁺ CD8⁺ T cells. **J–O** Xenogeneic GvHD model. **J** Experimental setup for the xenogeneic transplantation. **K** Summary graph showing the population of PD-1⁺ cells among human CD8⁺ T cells. **L** Representative flow plots showing the population of PD-1⁺CD8⁺ T cells and their TCF1, Tim-3, and TOX expression. **M** Summary graph showing the population of the TCF1⁺ subset among PD-1⁺ CD8⁺ T cells. **N** Representative flow plots of the population of TCF1⁺PD-1⁺ and TCF1⁻PD-1⁺ CD8⁺ T cell subsets and their CD45RA and CCR7 expression. **O** Summary graph showing the composition of TCF1⁺PD-1⁺ and TCF1⁻PD-1⁺ CD8⁺ T cells regarding naïve T cells (Tn, CCR7⁺CD45RA⁺), central memory T cells (Tcm, CCR7⁺CD45RA⁻), effector memory/ effector T cells (Tem/eff, CCR7⁻CD45RA⁻), and CD45RA⁺ effector memory T cells (Temra, CCR7⁻CD45RA⁺). Data were obtained from a single experiment (n = 5) for each transplantation setup, and the mean and SEM were shown. Source data are provided as a Source Data file.

## Distinct transcriptional profiles of PD-1⁺ alloreactive CD8⁺ T cell subsets in acute GvHD

To understand the differentiation of alloreactive CD8⁺ T cells in greater detail, we isolated PD-1⁺ CD8⁺ T cells from GvHD B6 recipients of Balb/c T cells at 33–35 dpt and performed scRNA-seq for transcriptomic analysis. Sub-clustering of the PD-1⁺ CD8⁺ T cells identified 14 clusters (Fig. 5A) that were further combined into three main subsets: progenitor T cells (Prog_T), proliferating T cells (Prolif_T), and terminally differentiated T cells (Term_T) (Fig. 5B). We first confirmed that all three subsets highly expressed *Cd8a*, *Pdcd1*, and *Tox*, validating PD-1⁺ CD8⁺ T cells (Fig. 5C). In addition to *Tcf7*, progenitor T cells highly expressed *Slamf6* and *Il7r*, which are markers identifying progenitor CD8⁺ T cells in chronic viral infection and cancers[12–20]. Proliferating T cells were defined with high expression of *Mki67*, *Stmn1*, and *Pclaf*, which have been implicated in cell proliferation[25,30–33]. Both proliferating and terminally differentiated CD8⁺ T cell subsets expressed the effector molecule *Fasl*, inflammatory cytokine *Csf1*, and chemokine receptor *Ccr5*, but their expression was much higher in the terminally differentiated subset than proliferating T cells. We next compared gene signatures between acute GvHD and chronic LCMV infection. Importantly, gene set enrichment analysis revealed that progenitor,

proliferating, and terminally differentiated subsets in acute GvHD exhibited similar gene signatures to the corresponding subsets of exhausted CD8⁺ T cells, CD101⁻Tim-3⁻ progenitor exhausted, CD101⁻Tim-3⁺ transitory exhausted, and CD101⁺Tim-3⁺ terminally exhausted CD8⁺ T cells in chronically LCMV-infected mice (Fig. 5D). This is consistent with similar phenotypic characteristics of CD8⁺ T cell subsets determined by flow cytometry.

We next focused on the disparate differentiation between progenitor and terminally differentiated CD8⁺ T cell subsets. From the pair-wise analysis, we found that the progenitor CD8⁺ T cell subset showed high expression of transcription factors *Id3*, *Lef1*, and *Klf2* in addition to *Tcf7* (Fig. 5E). Furthermore, *Xcl1*, a chemokine that plays a role in recruiting XCR1⁺ CD8α⁺ lymphoid dendritic cells (DCs)[34], and *Emb*, a fibronectin receptor maintaining the stemness of progenitor cells[35], were also highly expressed in the progenitor subset. In contrast, the terminally differentiated CD8⁺ T cell subset exhibited high expression of effector molecules (*Gzmb*, *Prf1*, *Fasl*, *Nkg7*, and *Ifng*), inhibitory receptors (*Havcr2*, *Entpd1*, *Lag3*, and *Cd160*), and inflammatory chemokine and chemokine receptors (*Ccl3*, *Ccl4*, and *Ccl5*; *Ccr5* and *Cxcr6*). Additionally, *Prdm1* and *Id2* were reciprocally enriched in the terminally differentiated CD8⁺ T cell subset. Network analysis

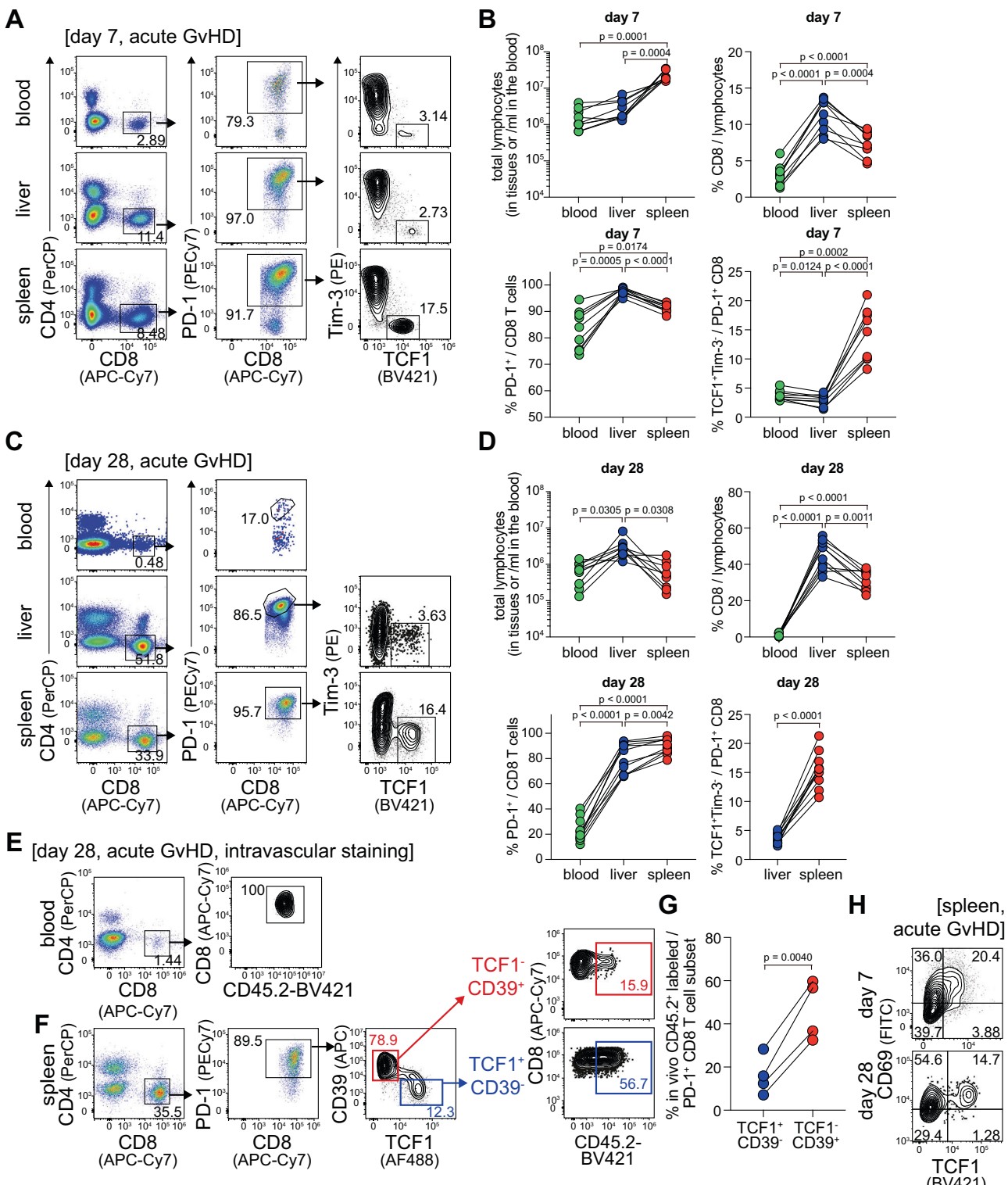

**Fig. 4 | Preferential localization of TCF1⁺PD-1⁺ CD8⁺ T cell subset in the spleen of mice with acute GvHD. A–D** Populations of CD8⁺, PD-1⁺ CD8⁺, and TCF1⁺PD-1⁺ CD8⁺ T cells in the blood, liver, and spleen of mice with acute GvHD at days 7 and 28 post-transplantation. Representative flow plots (**A**, **C**) and summary graphs (**B**, **D**) showing the frequency of CD8⁺, PD-1⁺ CD8⁺, and TCF1⁺PD-1⁺ CD8⁺ T cells and the absolute number of total lymphocytes in the tissues at days 7 (**A**, **B**) and 28 (**C**, **D**) post-transplantation. Data are combined from two independent experiments (*n* = 9 for day 7 and *n* = 10 for day 28). **E–G** In vivo CD45.2 staining was determined in the blood and spleen of mice with acute GvHD at 28 dpt, 3 min after injection. In vivo

CD45.2 labeling on circulating CD8⁺ T cells (**E**) and TCF1⁺CD39⁻ and TCF1⁻CD39⁺ PD-1⁺ CD8⁺ T cell subsets in the spleen (**F**). **G** Summary graph showing the frequency of in vivo CD45.2-labeled cells in each subset. Data are representative of two independent experiments (*n* = 4/experiment). **H** Expression of TCF1 vs. CD69 on PD-1⁺ CD8⁺ T cells in the spleen of mice with acute GvHD at the indicated time points. Statistical significance was determined by one-way ANOVA with post hoc Tukey's multiple comparisons test or a two-tailed paired *t*-test. Source data are provided as a Source Data file.

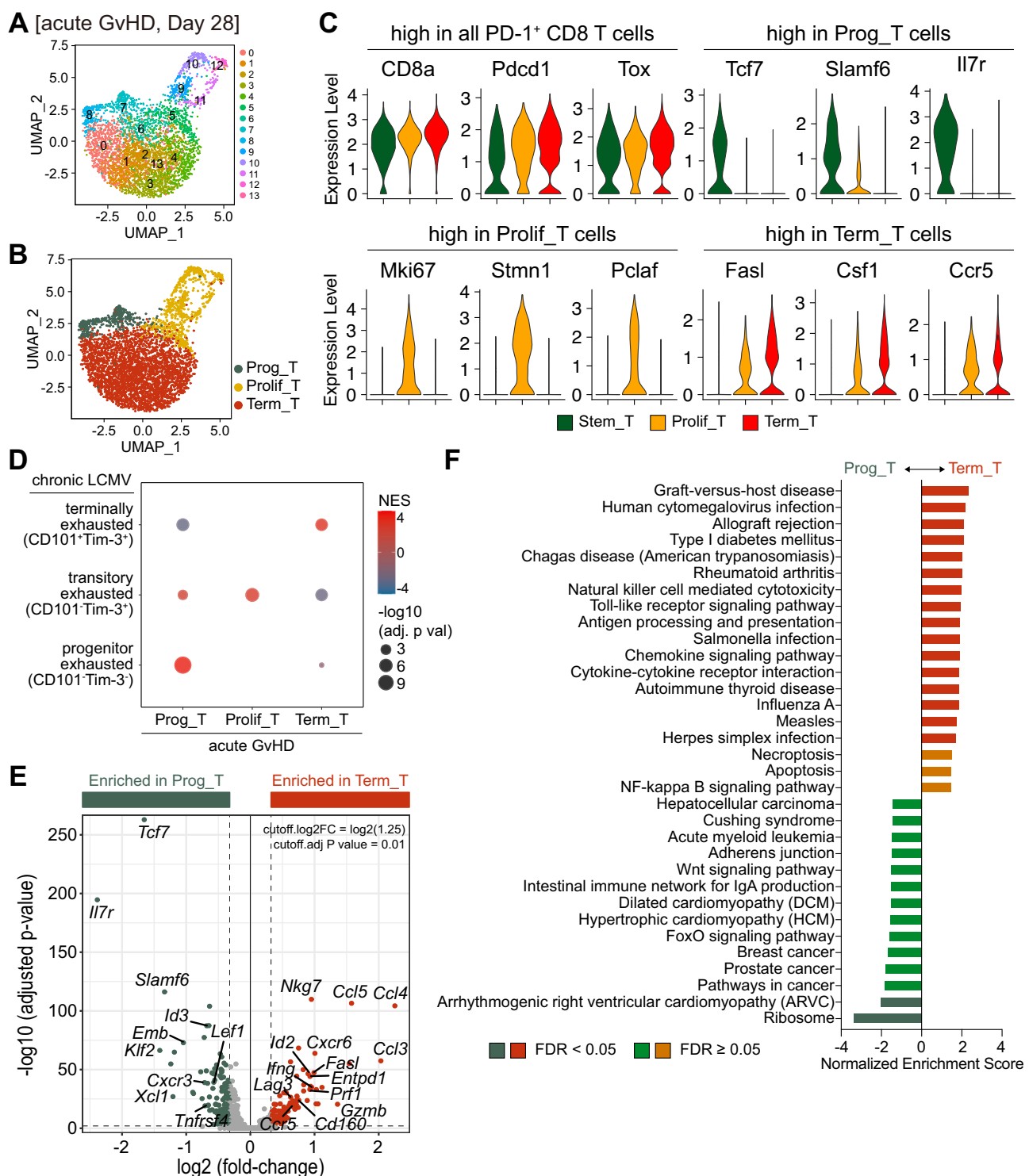

**Fig. 5 | Distinct transcriptional profiles of PD-1⁺ alloreactive CD8⁺ T cell subsets in acute GvHD.** ScRNA-seq was performed using PD-1⁺ alloreactive CD8⁺ T cells isolated from the spleen of mice with acute GvHD at 33–35 dpt. **A, B** UMAP showing unsupervised Seurat clusters (**A**) and supervised cluster groupings based on the phenotype (**B**) of PD-1⁺ CD8⁺ T cells across fourteen integrated samples. **C** Violin plots of representative genes highly expressed in all three PD-1⁺ CD8⁺ T cell subsets or differentially expressed in each subset. **D** Gene set enrichment analysis for comparing specific gene signatures of PD-1⁺ CD8⁺ T cell subsets in acute GvHD to those of exhausted CD8⁺ T cell subsets in chronically infected mice (available in PRJNA497086). The depth of the color represents the normalized enrichment score. The area of the circle in the graph reflects the adjusted *P*-value determined by one-tailed permutation test. **E** Comparison of differentially expressed genes between progenitor and terminally differentiated PD-1⁺ CD8⁺ T cell subsets. Volcano plot shows fold-change (log₂) versus adjusted *P*-value (-log₁₀) for each gene. Significance was determined as |log₂fold-change > log₂(1.25)| and adjusted *P*-value < 0.01, two-sided Wilcoxon test. **F** KEGG pathway analysis of the highly expressed genes in each subset. A false discovery rate (FDR) < 0.05 indicates significant change. Data are representative of 10 biologically independent pooled samples. Source data are provided as a Source Data file.

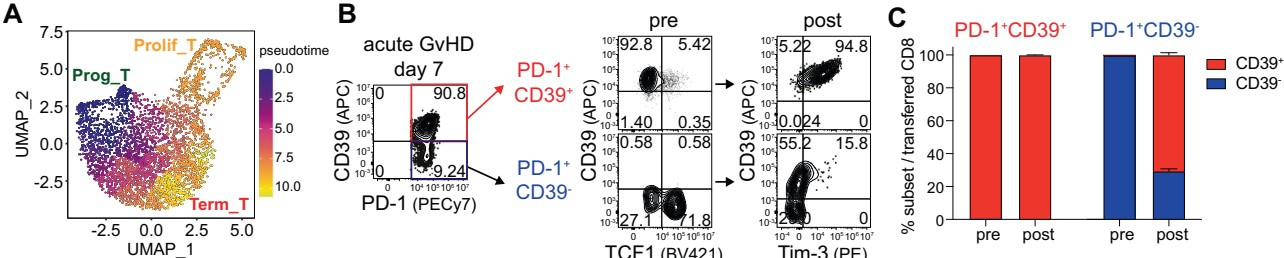

**Fig. 6 | Progenitor-progeny relationship of PD-1⁺ CD8⁺ T cell subsets in acute GvHD. A** Pseudotime analysis of PD-1⁺ alloreactive CD8⁺ T cells showing a differentiation trajectory where cells begin as progenitor cells (Prog_T), transition through proliferating cells (Prolif_T), and finally become terminally differentiated cells (Term_T). **B, C** PD-1⁺CD39⁻ cells containing a large fraction of TCF1⁺ cells and PD-1⁺CD39⁺ cells, mostly TCF1⁻, were isolated from the spleen of GvHD B6 recipients of Balb/c T cells at 7 dpt and co-cultured with naïve CD45.1⁺ B6 splenocytes in the presence of CD8⁺-depleted naïve Balb/c splenocytes for 5 days. **B, C** Phenotypic analysis (**B**) and summary graph (**C**) showing CD39⁺ and CD39⁻ proportions among sorted PD-1⁺CD39⁻ and PD-1⁺CD39⁺ CD8⁺ T cell subsets before and after the MLR. Data are representative of two similar experiments, $n = 3$ per experiment. Source data are provided as a Source Data file.

further indicated differential functional aspects of progenitor and terminally differentiated CD8⁺ T cell subsets. KEGG pathway analysis revealed that gene sets associated with GvHD were top-ranked in the terminally differentiated subsets (Fig. 5F). Furthermore, genes related to infections (human cytomegalovirus, Salmonella, Influenza A, measles, and herpes simplex), allograft rejection, and autoimmune diseases (type I diabetes mellitus, rheumatoid arthritis, and autoimmune thyroid disease) were enriched in the terminally differentiated subsets. Necroptosis- and apoptosis-associated genes were also enhanced in the terminally differentiated subsets. In contrast, ribosomal proteins were highly expressed in the progenitor subset. Furthermore, gene sets associated with the Wnt signaling pathway, FoxO signaling pathway, and cancers were enriched in the progenitor subset. The Reactome pathway also revealed that translation-related genes were enhanced in the progenitor subset, while the terminally differentiated subset highly expressed genes in DAP12 signaling (Supplementary Fig. 9), which contributes to the effector function of CD8⁺ T cells and NK cells[36]. Overall, these results suggest that alloreactive CD8⁺ T cells in acute GvHD are heterogeneous and comprise progenitor and effector-like CD8⁺ T cell subsets similar to the differentiation of exhausted CD8⁺ T cells in chronic LCMV infection.

**Exclusive proliferative potential of the TCF1⁺PD-1⁺ CD8⁺ T cell subset and its differentiation into TCF1⁻PD-1⁺ cells in acute GvHD**

To elucidate the lineage relationship of PD-1⁺ alloreactive CD8⁺ T cell subsets in their differentiation pathway, we performed pseudotime analysis. The analysis revealed that PD-1⁺ alloreactive CD8⁺ T cells started from the progenitor subset (Prog_T), progressed into the proliferating subset (Prolif_T), and then reached the terminally differentiated subset (Term_T) (Fig. 6A). Therefore, they lost the progenitor signatures while gaining effector-like signatures during differentiation. To verify their progenitor-progeny relationship directly, we isolated PD-1⁺CD39⁻ cells, of which ~70% were TCF1-positive, and PD-1⁺CD39⁺ cells, which were mostly TCF1-negative, from the spleen of GvHD B6 recipients of Balb/c T cells at 7 dpt (Fig. 6B). We opted for 7 dpt because it was difficult to obtain enough cells for analysis at 28 dpt (Fig. 2B). We then performed two-way MLR by incubating sorted CD8⁺ T cells with T cell–depleted Balb/c splenocytes and total CD45.1⁺ B6 splenocytes because one-way MLR was less effective in upregulating Tim-3 and CD39 on alloreactive CD8⁺ T cells (Supplementary Fig. 10). As illustrated in Fig. 6B, most sorted CD39⁺ sustained a CD39⁺Tim-3⁺ phenotype, reflecting their terminally differentiated state. In contrast, sorted CD39⁻ CD8⁺ T cells produced a large number of CD39⁺ cells after 5 days of incubation; therefore, more than half of the transferred cells became CD39-positive (Fig. 6B–C). Furthermore, a small fraction of the progeny CD39⁺ cells became CD39 and Tim-3 double-positive cells. Thus, these results confirmed that TCF1⁺ cells act

as progenitor cells supporting TCF1⁻ progeny cells through antigenic stimulation during acute GvHD.

We next explored the in vivo function of TCF1⁺ T cells using a two-stage GVHD model (Fig. 7A–C). We first developed acute GvHD by transferring splenocytes and bone marrow cells from CD45.2 B6 mice (H2ᵇ) into lethally irradiated Balb/c (H2ᵈ) mice (Fig. 7A). Seven days post-transplantation, we isolated PD-1⁺Tim-3⁻ cells, of which ~16% were TCF1⁺, and PD-1⁺Tim-3⁺ cells, which were mostly TCF1⁻, from the spleen of GvHD Balb/c recipients of CD45.2⁺ B6 T cells (Fig. 7B). Isolated H2ᵇCD45.2⁺ CD8⁺ cells were co-transferred into lethally irradiated Balb/c (H2ᵈ) mice with bone marrow cells obtained from CD45.1 B6 mice to distinguish sorted splenocytes and bone marrow cells (Fig. 7C). We sacrificed mice 14 dpt when the recipients started to die. First, we observed a higher frequency of transferred PD-1⁺Tim-3⁻ CD8⁺ T cells among total CD8⁺ T cells than PD-1⁺Tim-3⁺ cells in the peripheral tissues we examined. For example, the infused PD-1⁺Tim-3⁺ CD8⁺ T cells comprised ~0.7% of the total CD8⁺ T cells, while PD-1⁺Tim-3⁻ CD8⁺ T cells accounted for around 7% (Fig. 7D, E). Assuming that 10% of the transferred cells enter the spleen, we observed a 10-fold increase in infused PD-1⁺Tim-3⁻ CD8⁺ T cells, while the number of infused PD-1⁺Tim-3⁺ CD8⁺ T cells decreased by half (Fig. 7E). Next, we examined the phenotype of the infused cells in terms of the expression of Tim-3 and CD39. Consistent with the results of the ex vivo MLR study (Fig. 6B, C), the majority of transferred Tim-3⁺ cells maintained their original phenotype, whereas transferred Tim-3⁻ cells mostly differentiated into Tim-3⁺ cells (Fig. 7F, G). Collectively, these results suggest that PD-1⁺Tim-3⁻ cells possess exclusive proliferative potential and could differentiate into PD-1⁺Tim-3⁺ CD8⁺ T cells in vivo during acute GvHD.

**Impaired proliferative potential of PD-1⁺ alloreactive CD8⁺ T cells in acute GvHD**

In the current study, we found that alloreactive CD8⁺ T cells highly expressed PD-1 and Tox (Fig. 1 and Supplementary Figs. 2 and 5). These molecules are widely known to be upregulated in persistent antigenic stimulation and to be associated with T cell dysfunction in chronic viral infection and cancer[37–41]. Next, we examined whether the proliferative potential of alloreactive CD8⁺ T cells was diminished corresponding to their high PD-1 and Tox expression compared with naïve CD8⁺ T cells. Two-way MLR was performed by the incubation of naïve B6 splenocytes with (1) naïve Balb/c splenocytes, (2) GvHD Balb/c-derived splenocytes from B6 recipients (day 21 post transplantation), or (3) GvHD Balb/c-derived CD8⁺ T cells from B6 recipients (day 21 post transplantation) plus CD8⁺-depleted naïve Balb/c splenocytes (Fig. 8A). We first observed more proliferation in naïve B6 splenocytes than in naïve Balb/c splenocytes (Fig. 8B, C), suggesting the higher frequency of alloreactive CD8⁺ T cell precursors to the other strain in B6 mice. When incubated with GvHD Balb/c-derived splenocytes from B6 recipients, the proliferation of naïve B6 splenocytes was completely

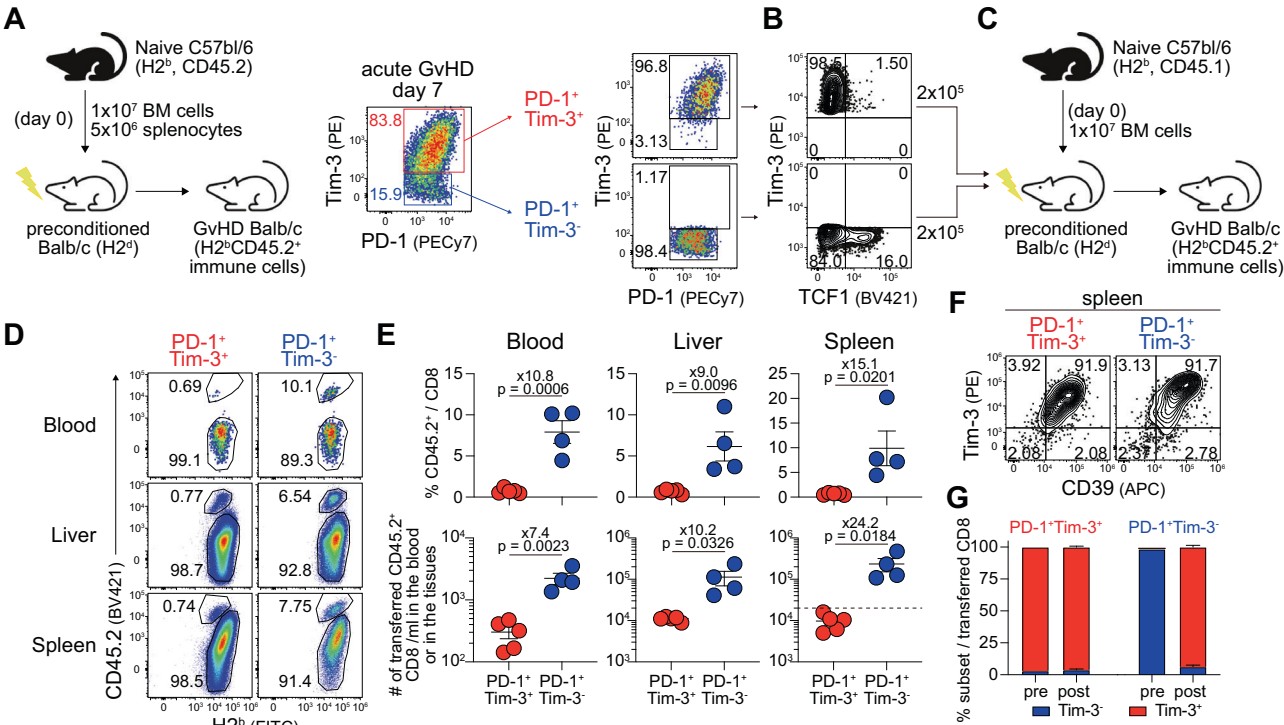

**Fig. 7 | In vivo exclusive proliferative potential of the TCF1+ CD8+ T cell subset and their differentiation into Tim-3+ cells upon acute GvHD. A** Experimental setup (1): PD-1+Tim-3+ and PD-1+Tim-3- CD8+ T cells were isolated from the spleen of GvHD Balb/c recipients of B6 T cells at 7 dpt. **B** Population of TCF1+ cells among sorted PD-1+ CD8+ T cells. **C** Experimental setup (2): Sorted CD8+ T cells (2×10⁵) were transferred into lethally irradiated Balb/c mice with 1×10⁷ bone marrow cells intravenously. **D, E** Representative flow plots (**D**) and summary graphs (**E**, top) showing the frequency of sorted CD45.2+ cells among CD8+ T cells and their absolute number (**E**, bottom) in the indicated tissues. A dashed line indicates the number of transferred cells, assuming a 10% engraftment rate. **F, G** Phenotypic analysis (**F**) and summary graph (**G**) showing the proportions of Tim-3+ and Tim-3- within sorted CD8+ T cell subsets in the spleen at 7 dpt compared to the pre-transplantation state. Data were obtained from a single experiment (n = 5 for PD-1+Tim-3+ cells and n = 4 for PD-1+Tim-3- cells), and the mean and SEM were shown. Statistical significance was determined by a two-tailed unpaired *t*-test. Source data are provided as a Source Data file.

abrogated whereas it was restored by supplementation with CD8+-depleted naïve Balb/c splenocytes, supporting the notion that allogeneic antigen-presenting cells (APCs) are essential for the activation of alloreactive CD8+ T cells, and that APCs in mice with acute GvHD were functionally debilitated. In the presence of CD8+-depleted naïve Balb/c splenocytes and naïve B6 splenocytes, ~30% of GvHD Balb/c-derived CD8+ T cells from B6 recipients exhibited alloreactive proliferation. However, more than half of carboxyfluorescein diacetate succinimidyl ester (CFSE)-diluted cells achieved only one division and fewer than 3 divisions were observed (Fig. 8D, E). In contrast, naïve Balb/c splenocytes showed up to 5 divisions and were evenly distributed at each division. Consistent results were also obtained from the experiment using GvHD Balb/c-derived CD8+ T cells from B6 recipients isolated on day 7 post transplantation (Supplementary Fig. 11). Taken together, these results suggest that although alloreactive PD-1+ CD8+ T cells possessed proliferative potential after antigenic stimulation, it was impaired compared with that of naïve alloreactive CD8+ T cells.

**IL-15-driven proliferation of PD-1+ alloreactive CD8+ T cell subsets in acute GvHD**
Similar to memory CD8+ T cells[42], we recently reported that the TCF1+ progenitor exhausted CD8+ T cell subset exclusively self-renewed after ex vivo and in vivo IL-15 stimulation in chronically LCMV-infected mice and patients with metastatic renal cell carcinoma[43]. We first observed that only TCF1- CD8+ T cells showed minimal CD122 (IL-2/15Rβ) expression, while both subsets exhibited considerable expression of CD132 (common ɣ chain receptor) at 28 days post allogeneic transplantation (Fig. 9A, B). To examine the IL-15-driven homeostatic

proliferation of PD-1+ alloreactive CD8+ T cell subsets, we isolated CD8+ T cells from the spleen of GvHD B6 recipients of Balb/c T cells at 28 dpt, labeled them with CFSE, and cultured them with 100 ng/ml of recombinant IL-15 protein for 3 days. Of note, both TCF1+CD39- and TCF1-CD39+ subsets diluted CFSE signals after ex vivo IL-15 treatment (Fig. 9C, D). However, IL-15-driven proliferation was still much higher in TCF1+CD39- cells (around 45%) than TCF1-CD39+ cells (~30%). Taken together, although both TCF1+ and TCF1- subsets proliferated considerably after IL-15 stimulation during acute GvHD, TCF1+PD-1+ CD8+ T cells were more responsive to IL-15 than TCF1-PD-1+ CD8+ T cells.

## Discussion
In this study, we comprehensively explored the heterogeneity of PD-1+ alloreactive CD8+ T cells using both flow cytometry analysis and scRNA-seq in acute GvHD driven by allogeneic transplantation. Although acute GvHD and chronic LCMV infection may appear to be completely different conditions, we hypothesized that there might be similarities in the CD8+ T cell differentiation program between these two models due to the persistent stimulation of a high antigen level. We did observe analogous differentiation features of CD8+ T cells between chronic LCMV infection and MHC-mismatched acute GvHD through direct comparison. Furthermore, the generation of the TCF1+PD-1+ CD8+ T cell subset was also verified in the models of MHC-matched, minor antigen-mismatched allogeneic GvHD and xenogeneic GvHD. Consistent with the results in chronic viral infection and cancer[12–20], TCF1+ progenitor CD8+ T cells possessed exclusive proliferative potential, were able to differentiate into TCF1- progeny CD8+ T cells harboring effector molecules via antigenic stimulation, and predominantly localized to the spleen. In addition to the recent

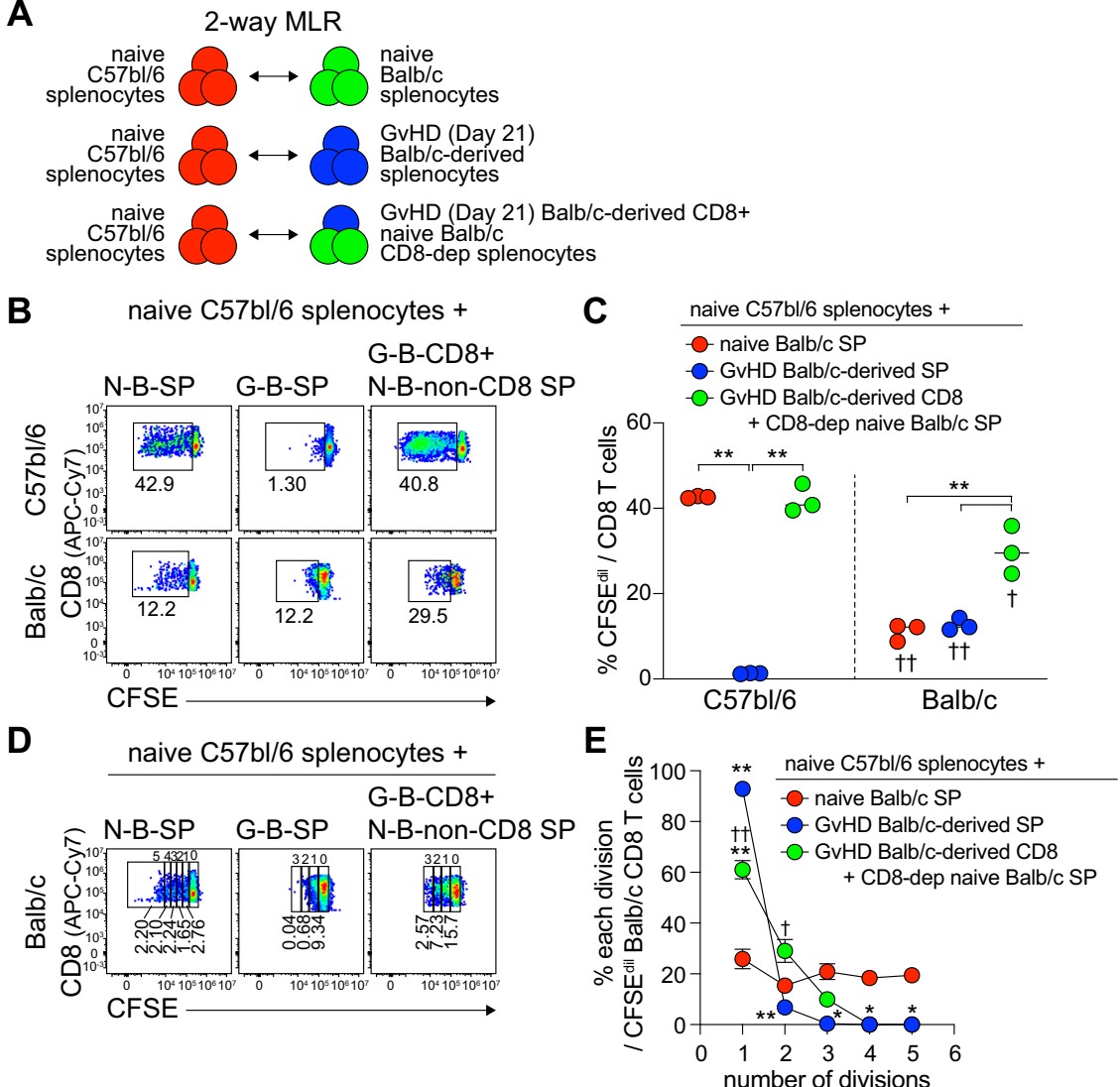

**Fig. 8 | Impaired proliferative potential of PD-1⁺ alloreactive CD8⁺ T cells in acute GvHD.** A Experimental setup: (1) Naïve Balb/c splenocytes ($2 \times 10^5$), (2) GvHD Balb/c-derived splenocytes ($2 \times 10^5$), and (3) GvHD Balb/c-derived CD8⁺ T cells ($4 \times 10^4$) plus CD8⁺-depleted naïve Balb/c splenocytes ($1.6 \times 10^5$) were co-cultured with naïve CD45.1⁺ B6 splenocytes ($2 \times 10^5$) for 5 days in two-way MLR. Splenocytes of GvHD B6 recipients of Balb/c T cells were isolated at 21 dpt. **B, C** Representative plots (**B**) and summary graph (**C**) showing the proportion of CFSE-diluted cells among CD8⁺ T cells in the indicated conditions. Data were obtained from a single experiment ($n = 3$ of technical replicates), and the mean and SEM were shown. Statistical significance was determined by one-way ANOVA with post hoc Tukey's multiple comparisons test and was only shown for comparison within each mouse strain or between the same groups. **D** Representative plots of the proportion of CD8⁺ T cells at each division among Balb/c-derived CD8⁺ T cells. **E** Summary graph showing the proportion of CD8⁺ T cells at each division among CFSE-diluted Balb/c-derived CD8⁺ T cells. Statistical significance was determined by one-way ANOVA with post hoc Tukey's multiple comparisons test. \*$p < 0.05$ and \*\*$p < 0.01$ from the comparison with the group stimulated with naïve Balb/c splenocytes (1, red); †$p < 0.05$ and ††$p < 0.01$ from the comparison with the group stimulated with GvHD Balb/c-derived splenocytes (2, blue). Source data are provided as a Source Data file.

discovery of the TCF1⁺ CD8⁺ T cell subset in the models of GvHD and Graft-versus-leukemia (GvL)[44,45], Harris et al. demonstrated that the loss of TCF1 in donor CD8⁺ T cells reduced the severity and persistence of GvHD symptoms in the MHC-mismatched GvHD model[46]. These findings suggest that the generation of the TCF1⁺ progenitor CD8⁺ T cell subset represents a common mechanism of CD8⁺ T cell differentiation, which plays a crucial role in sustaining antigen-specific CD8⁺ T cells in microenvironments with persistent antigenic stimulation, such as allo-HSCT.

In addition to the generation of the TCF1⁺PD-1⁺ CD8⁺ T cell subset, we observed that PD-1⁺ alloreactive CD8⁺ T cells are functionally impaired in proliferative responses and exhibit resident features in peripheral tissues. The presence of similar differentiation programs of CD8⁺ T cells between chronic viral infection and acute GvHD, as well as

the identification of a TCF1⁺ progenitor subset among PD-1⁺ alloreactive CD8⁺ T cells, allows for new interpretations of previous momentous observations in the model of acute GvHD. First, it is well established that host APCs play a pivotal role in initiating CD8⁺ T cell–dependent acute GvHD[47,48]. A small number of host CD11c⁺ DCs are enough to prime the activation and differentiation of alloreactive T cells. In contrast, donor APCs rarely contribute to provoking acute GvHD, which might be from inefficient cross-presentation of alloantigens by transplanted APCs. However, given that much less severe GvHD was observed in the recipients of MHC I⁻/⁻ bone marrow cells[49], donor APCs are still instrumental in GvHD pathogenesis, probably by maintaining and/or augmenting the pool of alloreactive CD8⁺ T cells primed by host APCs. A study using a model of chronic LCMV infection demonstrated that high antigen loads in the early phase of infection

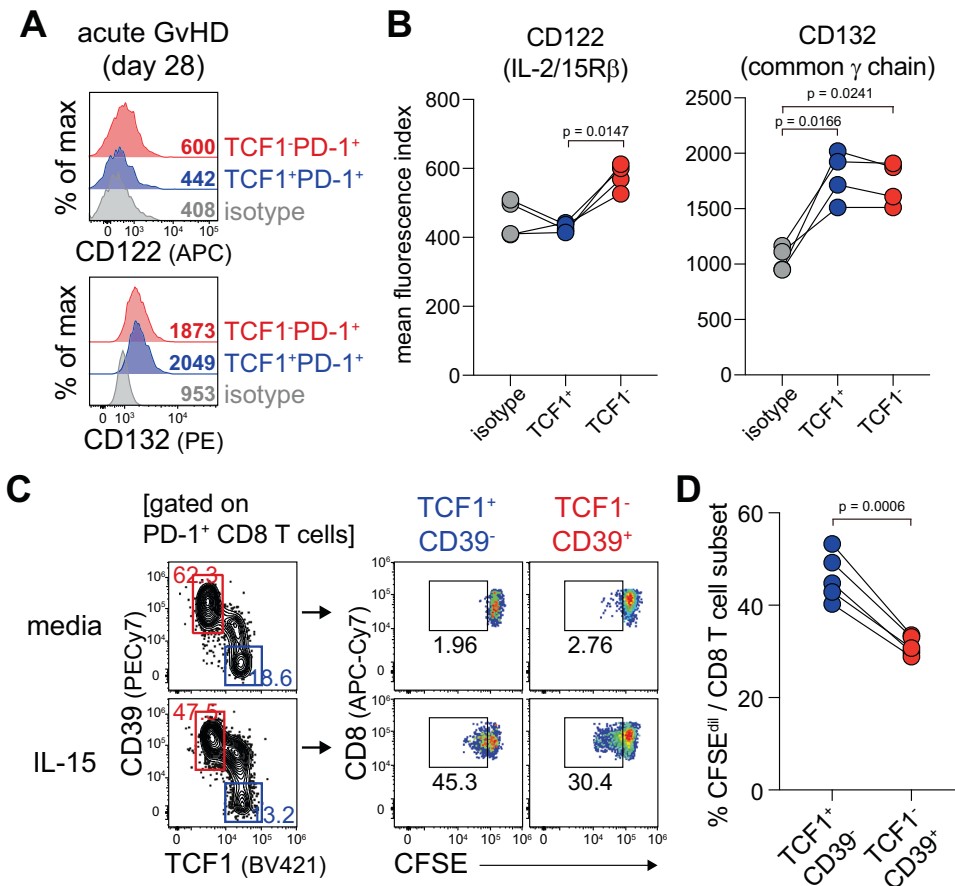

**Fig. 9 | Ex vivo IL-15-driven proliferation of the PD-1⁺ alloreactive CD8⁺ T cell subsets in acute GvHD. A**, **B** Representative flow plots (**A**) and summary graphs (**B**) showing the expression of CD122 (IL-2/15Rβ) and CD132 (common γ chain receptor) on the indicated CD8⁺ T cell subsets at 28 dpt. Data are representative of two independent experiments ($n = 4$/experiment). Statistical significance was determined by one-way ANOVA with post hoc Tukey's multiple comparisons test. **C**, **D** CD8⁺ T cells isolated from the spleen of GvHD B6 recipients of Balb/c T cells were labeled with CFSE and then cultured in the presence or absence of 100 ng/ml recombinant mouse IL-15 protein in addition to naïve Balb/c splenocytes for 3 days. **C** Gating strategy to determine TCF1⁺CD39⁻ and TCF1⁻ CD39⁺ subsets and their proliferation three days after the incubation. **D** Summary graph showing the proportion of CFSE-diluted cells among the indicated CD8⁺ T cell subsets. Data are representative of two similar experiments ($n = 4$ or 5/experiment). Statistical significance was determined by a two-tailed paired $t$-test. Source data are provided as a Source Data file.

led to the generation of a TCF1⁺ progenitor subset among virus-specific exhausted CD8⁺ T cells[50]. Similarly, abundant alloantigens presented on host APCs might promote the production of a TCF1⁺ PD-1⁺ alloreactive progenitor CD8⁺ T cell subset. Regarding the maintenance of the TCF1⁺ progenitor exhausted cells, Jansen et al. suggested that MHC II–expressing professional APCs provide a niche for the maintenance of progenitor exhausted CD8⁺ T cells in human kidney cancer[17]. A recent study in chronically infected mice also found that conventional type 1 DCs (cDC1s) act as a niche for sustaining progenitor exhausted CD8⁺ T cells via MHC-I-dependent interaction[51]. Although the reason why cross-presenting cDC1s, but not infected cDC2s and macrophages, play a role in sustaining the quiescence of progenitor exhausted CD8⁺ T cell subset is not fully understood, presenting lower levels of antigens through cross-presentation by cDC1 has been speculated to inhibit the transition/differentiation of the progenitor subset into effector-like cells. Therefore, it will be of interest to further investigate the role of DCs based on subsets in the pathogenesis of acute GvHD and their potential as a therapeutic target.

One of the major challenges in the allo-HSCT field is the separation of the GvL effect from the severity of GvHD. In addition to the identification of targets specific to hematopoietic cells[52], several strategies, including the treatment of tolerogenic anti-IL-2 antibodies and transient depletion of CD4 T cells, were examined for ameliorating GvHD while sustaining the GvL effect of allo-HSCT in patients with hematological malignancies[45,53]. However, given that most GvL effects were derived from alloreactive T cells instead of T cells specific to tumor-associated antigens (TAAs) or minor histocompatibility[54–56], it is more likely that the differentiation of T cells involved in GvHD and GvT would not be distinct but similar. In chronically infected mice, the first generation of early effector cells derived from naïve cells was polyfunctional and similar to effectors upon acute viral infection[50]. However, early effector cells were subjected to activation-induced cell death and the second generation of effector cells originated from the TCF1⁺ progenitor subset. Different from early effector cells, the second generation displayed compromised effector function. Of interest, in patients with hematologic malignancies received allo-HSCT, long-term disease-free survival was high when tumor cells were eradicated in the initial primary response[57,58], while T cell exhaustion occurred if residual cancer cells existed[59,60]. Therefore, after validation of the heterogeneity of alloreactive CD8⁺ T cells, further investigation is required to examine the function of effector T cells based on their origin (naïve vs. TCF1⁺ progenitor cells) and their role in the pathogenesis of GvHD and the induction of the GvL effect.

Another important question is which T cell subset in the transplantation is mainly responsible for the induction of acute GvHD. Zhang et al. previously demonstrated that CD44ʰⁱ CD8⁺ T cells activated by DCs ex vivo were defective in generating acute GvHD due to their early apoptotic deletion despite comparable cytolytic activity[61].

In addition, although memory CD8[+] T cells isolated from unprimed donors contained alloreactive clones and possessed proliferative potential upon short-term culture, they failed to induce GvHD due to their impeded maintenance[62]. In the same way, the comparison analysis between naïve and memory CD8[+] T cells revealed that memory CD8[+] T cells were not as effective as naïve cells in controlling tumor growth after transfer into tumor-bearing mice[63]. Similarly, notwithstanding their better expansion upon acute viral infection, memory CD8[+] T cells rapidly disappeared following persistent LCMV infection, which is different from naïve CD8[+] T cells[64]. In the system of trackable donor CD8[+] T cells for the GvL, memory alloreactive CD8[+] T cells produced a lower frequency of TCF1[+] cells than naïve CD8[+] T cells[44]. It has been proposed that this lower TCF1[+] population might be the reason why memory alloreactive CD8[+] T cells cause minimal GvHD even in H60 ubiquitously expressed transgenic mice. Together, these results suggest that more differentiated cells fail to generate the TCF1[+] progenitor CD8[+] T cell subset in the context of persistent antigenic stimulation, leading to a loss of antigen-specific T cells. This decrease in the population of effector cells results in tumor growth and the failure of viral control, but it also mitigates the symptoms of GvHD.

One limitation of our study was that PD-1 was used as a surrogate marker to determine alloreactive CD8[+] T cells instead of tetramer staining in all assays. Considering the increasing frequency of PD-1[+] cells among CD8[+] T cells in the order of syngeneic transplantation, MHC-matched/minor antigen-mismatched allogeneic transplantation, and MHC-mismatched allogeneic transplantation, it is plausible that PD-1[+] CD8[+] T cells were mostly alloreactive. However, the lack of tetramer staining still made it difficult to investigate the population of alloreactive CD8[+] T cells in the blood. Nevertheless, we observed a lower frequency of circulating CD8[+] T cells compared with other tissues at 28 dpt. Furthermore, in contrast to the spleen and liver in which a large fraction of CD8[+] T cells expressed PD-1, only 23% of CD8[+] T cells in the blood expressed PD-1. We also discovered a low frequency of labeled cells in the spleen after in vivo intravascular staining and high CD69 expression of splenocytes. Consistent with these findings, using the parabiosis technique, we verified the impaired circulating capability and residency of virus-specific exhausted CD8[+] T cells in chronically LCMV-infected mice, whereas antigen-specific memory CD8[+] T cells actively circulated between acutely infected parabionts[23]. In addition, tumor-specific tetramer-positive CD8[+] T cells were barely observed in the blood of patients with head and neck cancer, whereas a large number of PD-1[hi] tetramer-positive cells were found in the primary tumor and metastatic lymph nodes[16]. The role of CD69[+] resident memory T cells in the effector tissues in the development of GvHD has also been suggested[65,66]. Overall, these results indicate that PD-1[+] alloreactive T cells interact persistently with chronic antigens and become resident in the peripheral tissues where antigens exist. Further validation is required using a method to determine alloantigen-specific CD8[+] T cells.

In this study, we defined a TCF1[+]PD-1[+] progenitor population that could foster progeny cells producing effector molecules. Combined with the recent observation that TCF1 expression in CD8[+] T cells is essential in the pathogenesis of acute GvHD[46], this provides promising insight into opportunities for the development of therapeutics for acute GvHD. Eliminating the progenitor population or inhibiting the differentiation of the progenitor subset could be useful in mitigating acute GvHD. Our findings also offer a strategy for maximizing the differentiation of primary effectors while impeding the generation of the progenitor subset to augment the GvL effect of allo-HSCT without severe GvHD.

## Methods

### Mice, infection, transplantation and acute GvHD induction

Six- to eight-week-old female C57bl/6 (B6, H2[b]) and Balb/c (H2[d]) mice were purchased from Orient Bio (Gyeonggi, Republic of Korea). Eight-

week-old female 129/Sv (H2[b]) mice were purchased from DBL Co. (Stock #. 129SVE-F (Taconic), Chungbuk, Republic of Korea). CD45.1[+] B6 mice[67] were kindly provided by Dr. S. J. Ha (Yonsei University, Seoul, Republic of Korea), and we used eight-week-old female CD45.1[+] B6 mice and CD45.1[+]/CD45.2[+] B6 mice for experiments. The mice were kept in a controlled environment with a temperature of 23 °C and humidity ranging from 40% to 60%, following a 12-h light and 12-h dark, in a specific-pathogen-free (SPF) facility. After irradiation, the mice were transferred to and maintained in a semi-SPF facility. All experiments were conducted in accordance with the Institutional Animal Care and Use Committee guidelines of the Sungkyunkwan University School of Medicine.

For chronic LCMV infection, mice were infected with $4 \times 10^6$ plaque-forming units of LCMV clone 13 intravenously (i.v.). Transient CD4[+] T cell depletion was achieved by injecting 200 μg of anti-CD4[+] depleting antibodies (GK1.5, BioXCell, Lebanon, NH, USA) intraperitoneally (i.p.) two days before and on the day of infection to induce lifelong systemic infection with high levels of viremia[24].

For acute GvHD, mice underwent allogeneic bone marrow and splenocyte transplantation as previously described[68]. Briefly, recipient B6 mice were irradiated at a dose of 900 cGy in two fractions with a 2-h interval from a [137]Cs source. One day after the irradiation, $1 \times 10^7$ bone marrow cells and $4 \times 10^7$ splenocytes isolated from Balb/c or 129/Sv mice were transferred into irradiated B6 mice i.v. through the tail vein. For the syngeneic transplantation experiment, the recipient CD45.2 B6 mice were irradiated as described above. One day after irradiation, $1 \times 10^7$ CD45.1/CD45.2 bone marrow cells and $4 \times 10^7$ CD45.1 splenocytes isolated from congenically distinct B6 mice were transferred into irradiated CD45.2 B6 mice i.v. For the adoptive transfer experiment, recipient Balb/c mice were irradiated at a dose of 800 cGy in two fractions at a 2-h interval. One day after the irradiation, $1 \times 10^7$ bone marrow cells and $5 \times 10^6$ splenocytes or $2 \times 10^5$ sorted CD8[+] T cells isolated from naïve B6 or GvHD B6 mice, respectively, were transferred into irradiated Balb/c mice i.v.

We weighed recipient mice twice per week and monitored survival and the clinical score of acute GvHD as described previously[69]. The score was determined by five parameters: weight loss, posture, activity, fur texture, and skin integrity using a scale of 0 to 2, with 0 for absent or normal and 2 for severely abnormal, except for the body weight loss (score 0: <5%, 1: 5–10%, 2: 10–20%, and 3: >20%). The GvHD clinical index was determined at the sum of scores for each parameter (maximum = 11).

### Induction of xenogeneic GvHD

Eight-week-old male NSG mice were purchased from Ja Bio (Stock #. GEM-0022, Gyeonggi, Republic of Korea). Voluntarily donated peripheral blood from healthy individuals was supplied by Geonggi Blood Center, affiliated with the Korean Red Cross. The protocol was approved by the Institutional Review Board at Sungkyunkwan University School of Medicine (approval number: 2022-08-025-001). hPBMCs were isolated from whole blood using density gradient centrifugation with Histopaque-1077 (Sigma-Aldrich, Burlington, MA, USA). Briefly, blood samples were collected and diluted with an equal volume of phosphate buffered saline. Histopaque-1077 was gently underlaid beneath the diluted blood and centrifuged at 805 x g for 20 min at room temperature to perform density gradient centrifugation. To induce xenogeneic GvHD, recipient NSG mice were irradiated with a dose of 150 cGy in a single fraction. One day after irradiation, $5 \times 10^6$ hPBMCs were transferred into irradiated NSG mice i.v.

### Flow cytometry and in vivo intravascular staining

Lymphocytes were isolated from multiple tissues and blood as described previously[70]. All antibodies were purchased from BD Biosciences (San Jose, CA, USA), BioLegend (San Diego, CA, USA), Invitrogen (Carlsbad, CA, USA), or Cell Signaling Technology (Danvers, MA,

USA) (Supplementary Table 1). Major histocompatibility complex class I tetramers were prepared and used as previously described[71]. Transcription factors were stained using a Foxp3 transcription factor staining buffer set (Invitrogen). For in vivo antibody labeling, 3 μg of BV421-conjugated anti-CD45.1 antibody (BioLegend) was i.v. injected into mice with acute GvHD at 28 dpt. Peripheral blood mononuclear cells and splenocytes were isolated and used for direct ex vivo staining 3 min after the injection as described previously[29]. Dead cells were excluded using Live/Dead fixable dead cell stain kits (Invitrogen). All data were acquired on a CytoFLEX flow cytometer (Beckman Coulter, Brea, CA, USA) and analyzed using FlowJo software (Tree Star, Ashland, OR, USA).

### Cell sorting

Cell sorting was performed on a FACS Aria III Flow cytometer (BD Bioscience) of the BIORP. For single-cell RNA sequencing (scRNA-seq), PD-1[+] CD8[+] T cells were sorted from the spleen of GvHD B6 recipients of Balb/c T cells at 33–35 dpt. For ex vivo mixed lymphocyte reaction (MLR), PD-1[+] CD8[+] T cells, CD39[+] CD8[+] T cells, and CD39[-] CD8[+] T cells were sorted from the spleen of GvHD B6 recipients of Balb/c T cells at 7 dpt to >96% purity. CD8[+]-depleted Balb/c splenocytes were isolated from naïve mice. Magnetic associated cell sorting (MACS, Miltenyi Biotech, San Diego, CA, USA) was used to isolate total CD8[+] T cells from mice with acute GvHD and CD8[+]-depleted splenocytes from naïve Balb/c mice. For the adoptive transfer experiment, PD-1[+]Tim-3[+]H2[b] CD8[+] T cells and PD-1[+]Tim-3[-]H2[b] CD8[+] T cells were sorted from the spleen of GvHD Balb/c recipients of B6 T cells at 7 dpt with a purity >96%.

### ScRNA-seq

Libraries were prepared using the Chromium controller according to the 10x Chromium Next GEM Single Cell 3′ v3.1 protocol (CG000315). Briefly, the cell suspensions were diluted in nuclease-free water to achieve a targeted cell count of 10,000. The cell suspension was mixed with master mix and loaded with Single Cell 3′ v3.1 Gel Beads and Partitioning Oil into a Chromium Next GEM Chip G. RNA transcripts from single cells were uniquely barcoded and reverse-transcribed within droplets. cDNA molecules were pooled to go through an end repair process, the addition of a single 'A' base, and ligation of the adapters. The products were purified and enriched with PCR to create the final cDNA library. The purified libraries were quantified using qPCR according to the qPCR Quantification Protocol Guide (KAPA) and qualified using the Agilent Technologies 4200 TapeStation (Agilent Technologies). The libraries were sequenced using a HiSeq platform (Illumina) according to the read length in the user guide.

### Single-cell transcriptome data processing

The processed scRNA-seq data were aligned, filtered, and quantified using the Cell Ranger (v.5.0.0, 10x Genomics, Pleasanton, CA, USA) against the mm10 mouse reference genome. Further quality control and transcriptome analysis of the output data were performed using Seurat (v.4.0.6). Low-quality cells with either (1) < 2000 unique molecular identifiers or (2) >10% mitochondrial gene counts were filtered out. The sample object was processed using NormalizedData and FindVariableFeatures, then ScaleData, followed by RunPCA and RunUMAP. Louvain clustering was applied to identify each cell cluster.

### Differential expression analysis and trajectory analysis

All differentially expressed genes were identified using either FindAllMarkers or FindMarkers by a two-sided Wilcoxon test. The KEGG and Reactome pathway databases were applied in pathway analysis. A single-cell potential differentiation trajectory was constructed in R package Monocle3. Cells were ordered across the trajectory by tracking the changes in expression as pseudotime.

### Activation of alloreactive and xenoreactive CD8[+] T cells by MLRs

In MLR assays, splenocytes from naïve CD45.1[+] B6 mice were used to distinguish them from CD45.2[+] Balb/c splenocytes. For the conversion study, naïve CD45.1[+] B6 splenocytes and sorted CD8[+] T cells from GvHD B6 recipients of Balb/c T cells were labeled with CFSE (Invitrogen) to monitor lymphocyte proliferation. Then, $4 \times 10^4$ sorted CD8[+] T cells (H2[d]) from the GvHD B6 recipients of Balb/c T cells and $1.6 \times 10^5$ CD8[+]-depleted splenocytes isolated from naïve Balb/c (H2[d]) mice were co-cultured with $2 \times 10^5$ naïve CD45.1[+] B6 (H2[b]) splenocytes in 200 μl of culture medium in 96-well round-bottom plates for 5 days as a two-way MLR. For the comparison between one-way and two-way MLRs, $2 \times 10^5$ CFSE-labeled naïve CD45.2[+] Balb/c or CD45.1[+] B6 splenocytes (responders) were co-cultured with the same number of naïve splenocytes from the other strain with (one-way) or without (two-way) irradiations of 30 Gy as stimulators for 5 days. Responders were also incubated with splenocytes from the same strain (syngeneic) as a control. For xenogeneic MLR assay, hPBMCs from healthy subjects and splenocytes from NSG mice were utilized. hPBMCs were labeled with CFSE, and then $2 \times 10^5$ CFSE-labeled hPBMCs and $2 \times 10^5$ NSG splenocytes were co-cultured in 96-well round-bottom plates for 5 days in a two-way MLR setup.

### Ex vivo IL-15-driven proliferation assay

CD8[+] T cells were isolated from the spleen of GvHD B6 recipients of Balb/c T cells at 28 dpt. Cells ($2.5 \times 10^5$) were labeled with CFSE and mixed 1:2 with unlabeled naïve Balb/c splenocytes ($5 \times 10^5$) that were used as feeder cells. Cells were cultured in the presence and absence of recombinant mouse IL-15 (Peprotech, Rocky Hill, NJ, USA) at 100 ng/ml. After 3 days of culture, the cells were harvested and analyzed by flow cytometry.

### Statistical analysis

All statistical analyses were performed using Prism 8 software (GraphPad, San Diego, CA, USA). Statistical significance was determined by unpaired or paired two-tailed Student's $t$-test as indicated. $P$-values <0.05 or 0.01 were considered statistically significant. Statistical analysis from the scRNA-seq data was performed in R (v.4.1.2). The two-sided Wilcoxon test was used to compare specific gene or pathway signatures of PD-1[+] CD8[+] T cell subsets.

### Reporting summary

Further information on research design is available in the Nature Portfolio Reporting Summary linked to this article.

## Data availability

ScRNA-seq data are available at GEO under accession number (GSE215315). Each KEGG (Release 88.2, 11/01/2018) and Reactome (Version 66, September 2018) pathway database is available at https://www.genome.jp/kegg/pathway.html and https://reactome.org/PathwayBrowser/, respectively. Source data are provided with this paper.

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

## Acknowledgements

This study was supported by a National Research Foundation of Korea (NRF) grant (S.J.I.) funded by the Korean government (MSIT) (RS-2023-00211426 and 2023M3A9J405787312) and by the National Cancer Center, Korea (NCC-19112605) (S.J.I.). This research was also supported by Korea Basic Science Institute (National Research Facilities and Equipment Center) grant funded by the Ministry of Education (2020R1A6C101A191).

## Author contributions

S.J.I., S.L., and Ku.L. designed and performed the experiments and analyzed the data. H.B., Ky.L., J.L., and J.M. performed the experiments. Y.J.L., B.R.L., and W.-Y.P. analyzed single-cell RNA-seq data. S.J.I., S.L., and Ku.L. wrote the manuscript. All authors contributed to writing and provided feedback. The order of authors was determined by active and open discussion.

## Competing interests

The authors declare no competing interests.
