## [Peer Review File · Nature Communications]

Defining a TCF1-expressing progenitor allogeneic CD8 T cell subset in acute graft-versus-host diseaseREVIEWER COMMENTS

Reviewer #1 (Remarks to the Author):

CD8 T cells co-expressing TCF1 and PD-1 act as progenitor cells that ultimately give rise to antigen-specific effector cells. Presence of TCF1+ PD-1+ CD8 T cells has been associated with enhanced response to anti-PD-1 therapy. This underscores the biological and clinical relevance of these progenitor cells in cancer. In the present study, the authors outline the definition of a TCF-1-expressing progenitor subset in an acute GvHD mouse model. They found that TCF1+ and Tim-3- CD8 T cells were generated during the course of acute GvHD. The use of a mouse model of chronic viral infection as a control is very interesting, and the conclusions that the phenotype of TCF1+ PD-1+ cells is comparable in different disease contexts are providing important insights into CD8 T cell differentiation. Subsequent work indicates that in acute GvHD TCF1- cells stem from TCF1+ cells. TCF1 cells also respond better to IL-15 stimulation as compared to TCF-1 cells. Overall, the manuscript is well-written, and the results presented compelling and supportive. I have however two concerns that reduce my enthusiasm concerning this work, as follows:

- 1) One concern pertains to the model of GvHD used. As the authors are aware, multiple mouse models of GvHD have been described. The work performed in this particular model of acute GvHD is very informative from a mechanistic point of view. However, the results might not be sufficiently relevant to the human disease.
- 2) More importantly, while the phenotypic presentation of these TCF1+ PD-1+ cells in GvHD is comprehensive, the related in vivo functions of these cells have not been characterized.

Overall, the work presents a very nice and complete description of TCF1 PD-1 CD8 T cells in GvHD. Given the past studies published in other disease contexts, it is possible that these cells contribute to shape GvHD course. However, the relevance of these cells in GvHD remains insufficiently clear at this stage. This unfortunately reduces the impact of these investigations.

Reviewer #2 (Remarks to the Author):

The authors use flow cytometry and single cell RNA seq to describe heterogeneity of PD-1+ alloreactive CD8 T cells and identified a TCF1+ CD8 T cell subset. They demonstrate that TCF1 cells have lower expression of inhibitory receptors with higher costimulatory molecules and activation markers compared to TCF- cells. The TCF+ cells could become TCF- upon antigen stimulation. The allogeneic cells in the GVHD model became resident in the peripheral tissue and loss proliferative potential. The both the positive and negative fractions respond to IL-15 but the TCF+ proliferated more.

Comments:

The major concern this reviewer has is the difficulty with the comparison between GVHD and LCMV. In GVHD, there is a clear allogeneic response that is lacking in the LCMV response.

1. It seems that to assess LCMV within the first week of inoculation is too short to claim that this is a latent infection. That is, the timing of the assessment seems way too short to have confidence that the findings are significant as there is an initial pro inflammatory response before the virus becomes latent
2. The analysis of T cells in the allogeneic setting is difficult because it is not clear if the investigators are assessing only the donor T cells or what the contribution of the residual radiation resistant T cells are to their analysis
3. In both cases, the analysis within 3 days seems to be too short to then correlate with longer term outcomes.
4. In figure 1D, the similarities of chronic LCMV vs. GVHD on day 7 seems to be comparing apples vs. oranges. If day 7 is 7 days post LCMV infection, it would seem to be surprising that the same is in GVHD with a high PD1 given that the animal is at the height of their inflammatory response.
5. In figure 2 for GVHD studies, are these the donor or recipient cells?
6. In figure 3, it is surprising that one is able to find T cells of animals with GVHD on day 28, would be useful to know the absolute numbers given that the spleen in these animals are atrophic

Reviewer #3 (Remarks to the Author):

The manuscript by Lee et. al. investigated the differentiation of CD8 T cells associated with acute GVHD following MHC mismatched splenocyte and bone marrow transfer in mice. CD8 T cell differentiation was compared to CD8 T cells from chronic LCMV infected mice, although, it did not appear the LCMV-specific CD8 T cells were used. Rather, it appeared to be analysis of the total CD8 T cell population in LCMV infected mice. The CD8 T cell population in acute GVHD downregulated TCF1 and up-regulated Tim3 over the 28d study window, differentiating from stem-like memory cells to effector cells after proliferation. Consistent with their effector phenotype, TCF1⁻ cells proliferated less to IL-15 ex vivo than the TCF1⁺ population.

Overall, there is an extensive characterization of the memory T cell differentiation pathway after transplant, which appeared similar in phenotype to the total CD8 T cell population in chronic LCMV infection. In my opinion, this pathway would be anticipated based on the referenced literature and the value of knowing this pathway, and how it could be manipulated to mitigate acute GVHD, is not experimentally tested despite proposing multiple possibilities in the discussion. Therefore, I feel the manuscript reports mainly observational data of memory cell differentiation using a mouse model where extensive tools could be employed to provide strategies to improve acute GVHD. It does not mean the data are not relevant or important, but impact has to be considered.

Major comments

PD-1 is not only up-regulated by TCR stimulation. It can be up-regulated by cytokine exposure/proliferation. This is important because 90% of the cells appear to be activated in an allogeneic manner. Based on my experience in human MLR assays, the frequency of allo-specific cells is typically around 10%. Therefore, it is not clear if this is all due to allo-recognition by Cd8 T cells after transfer or homeostatic proliferation to fill the void left from irradiation. What does the profile of CD8 T cells look like after syngeneic transfer after lethal irradiation? Do they proliferate and show activation markers similar to the allogeneic situation in the paper?

No LCMV-specific CD8 T cell responses were measured in the study. The virus-specific CD8 T cells will represent a fraction of the activated CD8 T cell population and may demonstrate a different pathway of differentiation. Therefore, what is being measured is the impact of a persistent inflammatory environment on total CD8 T cell phenotype. This should be stated more clearly in the results and discussion.

We are grateful for your valuable and constructive suggestions, which have significantly enhanced the quality of our manuscript, entitled “Defining a TCF1-expressing progenitor allogeneic CD8 T cell subset in acute graft-versus-host disease.” We sincerely appreciate the time and effort you have dedicated to providing insightful feedback. In response to your comments, we have undertaken a series of new experiments to address the raised suggestions. We have diligently addressed each point and incorporated additional data, experimental responses, and clarifications to enhance the overall quality and clarity of our findings. Below, we provide a summary of the new experiments conducted, followed by our specific responses to your comments.

The following new experiments have been added to the revised version of our manuscript:

1. Figure 3A-E. We performed syngeneic transplantation by transferring B6 splenocytes and bone marrow cells into congenically distinct B6 recipient mice, as requested by Reviewer #3. Our findings indicate that upregulation of the PD-1 molecule was not mediated by proinflammatory cytokines or homeostatic proliferation of T cells in the lymphopenic state induced by conditioning.

2. Figure 3H-I. In response to a suggestion by Reviewer #1, we implemented a clinically relevant MHC-matched (H-2b), multiple minor antigen-mismatched model of bone marrow transplantation (129/Sv → B6) to enhance the clinical significance of our findings. Our initial observations revealed that the MHC-matched allogeneic GvHD model resulted in a lower frequency (~35%) of PD-1+ cells among CD8 T cells compared to the MHC-mismatched allogeneic GvHD model (~90%, Balb/c → B6). However, despite this difference, we were still able to identify a distinct population of TCF1+Tim-3-PD-1+TOX+ CD8 T cells, providing support for a shared differentiation program of CD8 T cells into the TCF1+PD-1+ subset during acute GvHD.

3. Figure 3J-O. To further enhance the clinical relevance of our findings, we next employed xenogeneic transplantation by transferring human PBMCs into NSG mice. Similar to the results in the MHC-mismatched GvHD model, our initial findings demonstrated that approximately 90% of infused human CD8 T cells expressed PD-1. Furthermore, around 15% of PD-1+ CD8 T cells exhibited the phenotype of TCF1+Tim-3-Tox+. These results further support the existence of a common differentiation program of CD8 T cells into the TCF1+PD-1+ subset in acute GvHD models.

4. Figure 7. In response to a suggestion by Reviewer #1, to verify the *in vivo* function of the TCF1+PD-1+ CD8 T cell subset, we isolated Tim-3-PD-1+ cells, which contain TCF1+PD-1+ cells, and Tim-3+PD-1+ CD8 T cells from the spleen of mice with acute GVHD. Subsequently, we transferred these subsets into lethally irradiated recipient mice along with bone marrow cells obtained from congenically distinct mice. Our initial findings indicate that the Tim-3-PD-1+ subset possesses proliferative potential upon exposure to allogeneic antigens and differentiates into Tim-3+PD-1+ progeny cells. These results suggest that Tim-3-TCF1+ PD-1+ cells serve as a resource to maintain allogeneic CD8 T cell immunity during the course of acute GvHD.

5. Supplementary Figure 3. As Reviewer #3 suggested, we evaluated the differentiation of LCMV-specific CD8 T cells using viral antigen-specific tetramers. Consistent with the notion that PD-1-expressing cells are predominantly LCMV-specific during chronic LCMV infection, our results demonstrate a similar differentiation pattern between LCMV-specific CD8 T cells and PD-1+ CD8 T cells.

Below are our specific responses to the comments of the three reviewers. The reviewers' comments are shown *in italics*.

Reviewer #1

CD8 T cells co-expressing TCF1 and PD-1 act as progenitor cells that ultimately give rise to antigen-specific effector cells. Presence of TCF1+ PD-1+ CD8 T cells has been associated with enhanced response to anti-PD-1 therapy. This underscores the biological and clinical relevance of these progenitor cells in cancer. In the present study, the authors outline the definition of a TCF-1-expressing progenitor subset in an acute GvHD mouse model. They found that TCF1+ and Tim-3- CD8 T cells were generated during the course of acute GvHD. The use of a mouse model of chronic viral infection as a control is very interesting, and the conclusions that the phenotype of TCF1+ PD-1+ cells is comparable in different disease contexts are providing important insights into CD8 T cell differentiation. Subsequent work indicates that in acute GvHD TCF1- cells stem from TCF1+ cells. TCF1 cells also respond better to IL-15 stimulation as compared to TCF-1 cells. Overall, the manuscript is well-written, and the results presented compelling and supportive. I have however two concerns that reduce my enthusiasm concerning this work, as follows:

1) *One concern pertains to the model of GvHD used. As the authors are aware, multiple mouse models of GvHD have been described. The work performed in this particular model of acute GvHD is very informative from a mechanistic point of view. However, the results might not be sufficiently relevant to the human disease.*

Response: To enhance the clinical significance of our findings, we further employed an MHC-matched (H-2b), multiple minor antigen-mismatched allogeneic GvHD model (129Sv to B6, Fig.3H-I) and a xenogeneic GvHD model (Fig.3J-O) in addition to the MHC-mismatched GvHD model. We identified a distinct population of TCF1+Tim-3- PD-1+ CD8 T cells and confirmed their high TOX expression. These results suggest that the generation of the TCF1+ progenitor CD8 T cell subset occurs as a shared differentiation program in this acute GvHD model.

2) *More importantly, while the phenotypic presentation of these TCF1+ PD-1+ cells in GvHD is comprehensive, the related in vivo functions of these cells have not been characterized.*

Response: We agree with the reviewer regarding the need to clarify the *in vivo* functions of these cells during acute GvHD. To address this concern, we performed a new experiment as described in Fig. 7. We observed an exclusive *in vivo* proliferative potential of Tim-3-(TCF1+) PD-1+ CD8 T cells in acute GvHD, which subsequently converted into Tim-3+PD-1+ cells. Additionally, in a recent study by Harris et al., the loss of TCF1 in donor CD8 T cells reduced the severity and persistence of GvHD symptoms in an MHC-mismatched GvHD model (Harris et al. (2023) Cancer Immunol Immunother [PMID: 36562825]). Taken together, these results suggest that the TCF1+PD-1+ CD8 T cell subset is important for maintaining allogeneic CD8 T cell immunity in acute GvHD. We have discussed this concern in the Discussion section.

Overall, the work presents a very nice and complete description of TCF1 PD-1 CD8 T cells in GvHD. Given the past studies published in other disease contexts, it is possible that these cells contribute to shape GvHD course. However, the relevance of these cells in GvHD remains insufficiently clear at this stage. This unfortunately reduces the impact of these investigations.

Response: We thank the reviewer for the constructive comments, and we hope that the new experimental data provided in our revision clarifies the relevance of these cells in acute GvHD.

Reviewer #2

The authors use flow cytometry and single cell RNA seq to describe heterogeneity of PD-1+ alloreactive CD8 T cells and identified a TCF1+ CD8 T cell subset. They demonstrate that TCF1

cells have lower expression of inhibitory receptors with higher costimulatory molecules and activation markers compared to TCF⁻ cells. The TCF⁺ cells could become TCF⁻ upon antigen stimulation. The allogeneic cells in the GVHD model became resident in the peripheral tissue and loss proliferative potential. The both the positive and negative fractions respond to IL-15 but the TCF⁺ proliferated more.

Comments:

The major concern this reviewer has is the difficulty with the comparison between GVHD and LCMV. In GVHD, there is a clear allogeneic response that is lacking in the LCMV response.

1. It seems that to assess LCMV within the first week of inoculation is too short to claim that this is a latent infection. That is, the timing of the assessment seems way too short to have confidence that the findings are significant as there is an initial pro inflammatory response before the virus becomes latent.

Response: First, we would like to clarify that LCMV Clone 13 strain results in chronic infection, not latent infection. Second, although chronic LCMV infection and acute GvHD may appear to be completely different situations, we hypothesized that there might be similarities in the CD8 T cell differentiation program between these two models due to the persistent stimulation of a high antigen dose. It has recently been reported that a high antigen load at the onset of chronic LCMV infection promotes the differentiation of the TCF1⁺ progenitor subset at early time points (Utzschneider et al. (2020) Nat Immunol [PMID: 32839610]). Based on this, we believe that comparing T cell characteristics at 7 days post-infection and post-transplantation was an appropriate approach to assess the similarity of CD8 T cell differentiation between these two models.

2. The analysis of T cells in the allogeneic setting is difficult because it is not clear if the investigators are assessing only the donor T cells or what the contribution of the residual radiation resistant T cells are to their analysis.

Response: We would like to thank the reviewer for pointing this out. In our new experiment using a syngeneic transplantation model, we found that only 3% of CD8 T cells were derived from host cells, while approximately 90% of the transplanted splenic CD8 T cells were donor-derived. Additionally, during acute GvHD, less than 1% of the total CD8 T cells were composed of host cells, as shown in new Supplementary Fig.1B. The lower frequency of host CD8 T cells in the tissues of mice with acute GvHD could be attributed to the killing of host cells by allogeneic donor

T cells. Therefore, the majority of T cells examined in the current study are donor T cells. We have addressed this issue in the revised manuscript in lines 114-116 and in Supplementary Fig. 1.

3. In both cases, the analysis within 3 days seems to be too short to then correlate with longer term outcomes.

Response: We agree with the reviewer's concern regarding the premature conclusions drawn from the analysis within 3 days and its correlation with long-term outcomes. However, as the main focus of this paper is not the impact of IL-15 on GvHD, but rather T cell differentiation in the context of acute GvHD, we feel that this specific issue regarding IL-15's effect on GvHD should be addressed in future studies. After careful consideration, we have decided to remove the relevant discussion from the manuscript.

4. In figure 1D, the similarities of chronic LCMV vs. GVHD on day 7 seems to be comparing apples vs. oranges. If day 7 is 7 days post LCMV infection, it would seem to be surprising that the same is in GVHD with a high PD1 given that the animal is at the height of their inflammatory response.

Response: First, we would like to clarify that in Figure 1D, day 7 refers to 7 days post LCMV infection. Similar to the results observed in the MHC-mismatched allogeneic GvHD model, approximately 90% of the infused human CD8 T cells into NSG mice also exhibited a PD-1+ phenotype. In contrast, in the MHC-matched, multiple minor antigen-mismatched allogeneic GvHD model (129/Sv to B6), the frequency of PD-1+ cells (~35%) was much lower than that in the MHC-mismatched allogeneic GvHD model. Additionally, in syngeneic mice receiving donor CD8 T cells, only 10% of these cells expressed PD-1. These findings suggest that the upregulation of PD-1 on donor CD8 T cells can be attributed to the high antigen load of allogeneic or xenogeneic antigens.

5. In figure 2 for GVHD studies, are these the donor or recipient cells?

Response: We apologize for the lack of clarity. As mentioned in our response to Reviewer #2-2, the T cells analyzed in Figure 2 are also predominantly donor T cells. We have addressed this issue and provided clarification with the data in the revised manuscript.

6. In figure 3, it is surprising that one is able to find T cells of animals with GVHD on day 28, would be useful to know the absolute numbers given that the spleen in these animals are atrophic.

Response: We appreciate the reviewer's comment and their interest in the absolute numbers of T cells in animals with GVHD on day 28. The absolute numbers of total splenocytes, total CD8 T cells, and PD-1+ CD8 T cells were indeed presented in the original manuscript, specifically in Supplementary Figure 2 (now referred to as Supplementary Figure 4). We apologize if this information was not immediately apparent or if there was any confusion regarding the presentation of the data. In the revised manuscript, we have made adjustments to clearly indicate the location of the absolute numbers to avoid any further misunderstanding.

Reviewer #3

The manuscript by Lee et. al. investigated the differentiation of CD8 T cells associated with acute GVHD following MHC mismatched splenocyte and bone marrow transfer in mice. CD8 T cell differentiation was compared to CD8 T cells from chronic LCMV infected mice, although, it did not appear the LCMV-specific CD8 T cells were used. Rather, it appeared to be analysis of the total CD8 T cell population in LCMV infected mice. The CD8 T cell population in acute GVHD downregulated TCF1 and up-regulated Tim3 over the 28d study window, differentiating from stem-like memory cells to effector cells after proliferation. Consistent with their effector phenotype, TCF1- cells proliferated less to IL-15 ex vivo than the TCF1+ population.

Overall, there is an extensive characterization of the memory T cell differentiation pathway after transplant, which appeared similar in phenotype to the total CD8 T cell population in chronic LCMV infection. In my opinion, this pathway would be anticipated based on the referenced literature and the value of knowing this pathway, and how it could be manipulated to mitigate acute GVHD, is not experimentally tested despite proposing multiple possibilities in the discussion. Therefore, I feel the manuscript reports mainly observational data of memory cell differentiation using a mouse model where extensive tools could be employed to provide strategies to improve acute GVHD. It does not mean the data are not relevant or important, but impact has to be considered.

Major comments

1. PD-1 is not only up-regulated by TCR stimulation. It can be up-regulated by cytokine exposure/proliferation. This is important because 90% of the cells appear to be activated in an allogeneic manner. Based on my experience in human MLR assays, the frequency of allo-specific cells is typically around 10%. Therefore, it is not clear if this is all due to allo-recognition by Cd8 T cells after transfer or homeostatic proliferation to fill the void left from irradiation. What does the

profile of CD8 T cells look like after syngeneic transfer after lethal irradiation? Do they proliferate and show activation markers similar to the allogeneic situation in the paper?

Response: We appreciate the reviewer's suggestion and have conducted an experiment to address the role of cytokine exposure and homeostatic proliferation in PD-1 upregulation. In the syngeneic transfer model after lethal irradiation, we examined the phenotype of donor splenic CD8 T cells (Fig.3A-E). In contrast to the results observed in MHC-mismatched allogeneic GvHD, where approximately 90% of CD8 T cells expressed PD-1, only 10% of CD8 T cells expressed PD-1 in the syngeneic transplantation model. Furthermore, the phenotype of CD8 T cells in the syngeneic transfer model resembled that of naïve T cells despite high Ki-67 expression. These findings suggest that PD-1 upregulation is primarily driven by allo-antigenic stimulation rather than cytokine exposure and homeostatic proliferation. We have included these results in the revised manuscript to provide a more comprehensive understanding of PD-1 regulation in different transplantation settings.

2. No LCMV-specific CD8 T cell responses were measured in the study. The virus-specific CD8 T cells will represent a fraction of the activated CD8 T cell population and may demonstrate a different pathway of differentiation. Therefore, what is being measured is the impact of a persistent inflammatory environment on total CD8 T cell phenotype. This should be stated more clearly in the results and discussion.

Response: We appreciate the reviewer's comment and their insight regarding the measurement of LCMV-specific CD8 T cell responses in our study. It is important to note that the PD-1+ CD8 T cells analyzed in our study predominantly represent virus-specific CD8 T cells in the context of chronic LCMV infection. To address this concern, we conducted additional experiments using tetramers to specifically examine the phenotype of virus-specific CD8 T cells. The results consistently showed that the phenotype of virus-specific CD8 T cells was similar to that of PD-1+ CD8 T cells. These findings have been incorporated into a new figure (Supplementary Figure 3) in the revised manuscript.

In conclusion, we would like to express our sincere gratitude for your valuable and constructive suggestions, which have played a pivotal role in enhancing the quality of our manuscript. The revisions we have made have substantially strengthened the scientific integrity and overall quality of the study. We are confident that our findings on the TCF1-expressing progenitor allogeneic CD8 T cell subset in acute GvHD will make a significant contribution to the field of hematopoietic

stem cell transplantation. We are hopeful that the revised manuscript meets the high standards and expectations of the journal, and we eagerly anticipate the final decision regarding its publication.

REVIEWERS' COMMENTS

Reviewer #1 (Remarks to the Author):

The authors have included additional in vivo data that strengthen the manuscript. The results are sound and now fully support the conclusions drawn. I however concur with Reviewer 3 that the manuscript reports mainly observational data of memory cell differentiation. The authors discuss lines 430-35 the relevance of eliminating these progenitor cells, but after examining the data presented, the link with therapy remains unfortunately insufficiently documented at this stage.

Reviewer #2 (Remarks to the Author):

all concerns satisfied, except point #2 from my review. the use of syngeneic transplant does not really help in determining whether most of the responses are from donor cells in the allogeneic setting, but for the purposes of this manuscript it is acceptable.

Reviewer #3 (Remarks to the Author):

Having read the other reviewer comments, they appear closer to the field and recognize the impact of this work. Therefore, my previous comment related to impact may not be accurate.

The authors have addressed my comments with the suggested experiments, particularly syngeneic transfer, and the data look good.

I have no further comments.

We would like to thank Reviewer #1 for the additional comment. Here are our responses to Reviewer #1's specific comment (our responses are given in *italics*):

Reviewer #1.

Comments:

The authors have included additional in vivo data that strengthen the manuscript. The results are sound and now fully support the conclusions drawn. I however concur with Reviewer 3 that the manuscript reports mainly observational data of memory cell differentiation. The authors discuss lines 430-35 the relevance of eliminating these progenitor cells, but after examining the data presented, the link with therapy remains unfortunately insufficiently documented at this stage.

Response: We understand the reviewer's concern regarding the relevance of eliminating these progenitor cells for curing acute GvHD. During the revision, Harris et al. presented that the loss of TCF1 in donor CD8 T cells reduced the severity and persistence of GvHD symptoms in an MHC-mismatched GvHD model (Harris et al. (2023) Cancer Immunol Immunother [PMID: 36562825]). This result supports the idea that eliminating the TCF1+ CD8 T cell subset would be effective in treating acute GvHD. We have added this reference to the Discussion section as mentioned by the reviewer.

Reviewer #2.

Comments:

All concerns satisfied, except point #2 from my review. the use of syngeneic transplant does not really help in determining whether most of the responses are from donor cells in the allogeneic setting, but for the purposes of this manuscript it is acceptable.

Response: Thanks. We would like to inform the reviewer that more than 99% of CD8 T cells in the allogeneic setting were derived from donor cells, and the data are shown in Supplementary Figure 1.

Reviewer #3.

Comments:

Having read the other reviewer comments, they appear closer to the field and recognize the impact of this work. Therefore, my previous comment related to impact may not be accurate.

The authors have addressed my comments with the suggested experiments, particularly syngeneic transfer, and the data look good.

I have no further comments.

Response: *Thank you.*

In conclusion, we would like to express our sincere gratitude again for your valuable and constructive suggestions. We are hopeful that the revised manuscript meets the high standards and expectations of the journal, and we eagerly anticipate the final decision regarding its publication.